# Reasoning Can Be Restored by Correcting a Few Decision Tokens

Changshuo Shen [* 1]  Leheng Sheng [* 2]  Yuxin Chen [2]  Xiang Wang [1]  An Zhang [1]

## Abstract

Large reasoning models (LRMs) substantially outperform their base LLM counterparts on challenging reasoning benchmarks, yet it remains poorly understood where base models go wrong during token-by-token generation and how to narrow this gap efficiently. We study the base–reasoning gap through quantifying token-level distributional disagreement between a base model and a stronger reasoning model using likelihood-based divergences. Across benchmarks, we find that *the reasoning advantage is highly sparse and concentrates on a small set of early, planning-related decision tokens*. For instance, on Qwen3-0.6B, only ∼8% of generated tokens account for the salient disagreement, and these tokens concentrate early in the response, are strongly enriched in planning-related decisions (17×), and coincide with high base-model uncertainty—suggesting that base models fail mainly at early planning points that steer the subsequent reasoning trajectory. Building on this findings, we propose disagreement-guided token intervention, a simple inference-time delegation scheme that performs a one-token takeover by the reasoning model only at high-disagreement positions and immediately switches back to the base model. With a small intervention budget, this sparse delegation substantially recovers and can even surpass the performance of a same-size reasoning model on challenging reasoning tasks. Code is available at https://github.com/AlphaLab-USTC/RRTokenIntervention.

## 1. Introduction

[*]Equal contribution [1]University of Science and Technology of China, Hefei, China [2]National University of Singapore, Singapore. Correspondence to: An Zhang <an_zhang@ustc.edu.cn>.

*Proceedings of the 43rd International Conference on Machine Learning*, Seoul, South Korea. PMLR 306, 2026. Copyright 2026 by the author(s).

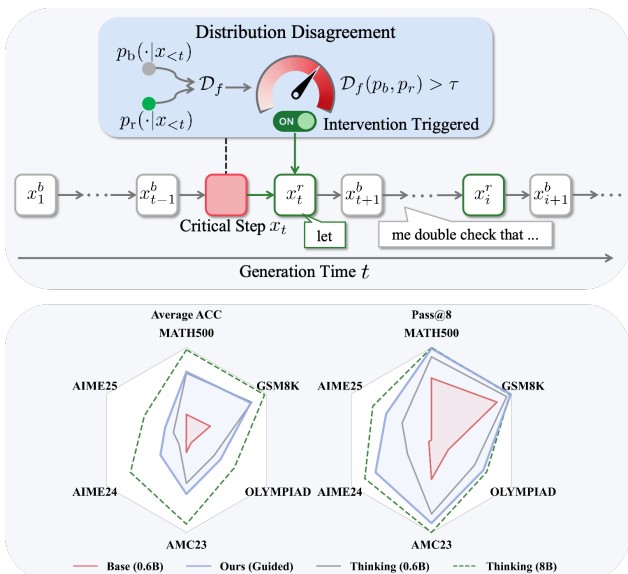

*Figure 1.* Disagreement-guided token intervention and its effectiveness. Intervening on 8% critical tokens, triggered by high token-level distributional disagreement through a strong LRM (*i.e.,* top figure), enables a 0.6B base model to recover most reasoning capabilities of the 8B strong LRM (*i.e.,* bottom figure) and outperform its post-trained 0.6B variant.

Large reasoning models (LRMs) have recently demonstrated substantial advantages over their base LLM counterparts on challenging reasoning benchmarks (Zhao et al., 2023; Jaech et al., 2024; OpenAI, 2024; Guo et al., 2025; Mathematical Association of America, 2024; He et al., 2024), with gains commonly attributed to test-time scaling (Li et al., 2025; Snell et al., 2025; Wang et al., 2023; Yao et al., 2023a) and reinforcement learning-based post-training such as RLVR (Shao et al., 2024; Wen et al., 2025). Beyond further improving performance, this widening base–LRM gap has motivated growing interest in a more fundamental question: *what* differs between base and reasoning models, and *why* does "reasoning mode" work? A growing body of evidence supports a latent-capability view in which base models may already encode considerable reasoning machinery, while post-training mainly activates, amplifies, or stabilizes it through minimal additional signals—such as activation-vector steering (Turner et al., 2025; Venhoff et al., 2025; Zou et al., 2023; Ward et al., 2025), confidence-based self-rewarding (Yuan et al., 2024; He et al., 2025),

distribution-level output amplification (Zhao et al., 2025), or even one-example unlocks (Wang et al., 2025b).

To address the *why* question, prior work has proposed hypotheses at multiple levels. Reasoning capabilities may emerge from eliciting richer intermediate trajectories, as in chain-of-thought prompting (Wei et al., 2022; Zhou et al., 2023; Wang et al., 2023); from inducing reasoning micro-behaviors such as backtracking (Yao et al., 2023a;b), verification (Lightman et al., 2024), and self-correction (Madaan et al., 2023; Pan et al., 2024; Marjanovic et al., 2025); from steering reasoning direction by restricting policy gradient updates to high-entropy tokens (Wang et al., 2025a); or from updating only a small subset of parameters, *i.e.,* a top-singular reasoning subspace (Cai et al., 2025; Hu et al., 2021). Despite these insights, we still lack a simple token-level account of *why* "reasoning mode" works—namely, where reasoning ability actually manifests during token generation and which decision points truly drive the base–LRM performance gap.

In this paper, we address this question by quantifying the base–LRM gap through token-level distributional disagreement between a base model and a stronger reasoning model using likelihood-based divergences such as cross-entropy and reverse Kullback–Leibler divergence (Cover & Thomas, 2001). We then characterize which tokens carry the reasoning advantage, how many such tokens exist, and what properties they exhibit. Across benchmarks, we find that reasoning ability is concentrated in a surprisingly small set of planning-related decision tokens that more likely appear early in the response. Concretely, only a small fraction of generated tokens determines whether the model commits to a reasoning trajectory: on Qwen3-0.6B (Yang et al., 2025), merely ∼8% of output tokens account for the salient base–LRM disagreement (*i.e.,* disagreement spike) (*cf.* Figure 1), indicating that token-level disagreement is highly sparse. Moreover, these disagreement spikes appear disproportionately early in generation (*cf.* Figure 2), contrary to the common intuition that reasoning gains primarily stem from long traces after the reasoning process unfolds. Finally, these critical tokens are tightly linked to planning under uncertainty: during recovery from base to reasoning roll-outs, planning-related tokens become markedly more likely, increasing by $17\times$ in occurrence probability (*cf.* Table 5), and they correlate strongly with the base model's intrinsic uncertainty (*cf.* Figure 2c). This suggests that base models struggle on complex reasoning tasks primarily at the planning points—where they are least confident are precisely the planning decisions that steer the subsequent reasoning trajectory.

Motivated by these findings, we ask whether a base model can attain strong reasoning performance with limited assistance from a reasoning model—by delegating only a small number of critical decisions. We propose a simple inference-time mechanism as shown in Figure 1: at positions where the base–LRM disagreement score is large, we perform a one-token takeover—the reasoning model generates only the next token at that position, after which generation immediately switches back to the base model (Ong et al., 2025; Huang et al., 2026). We term this sparse delegation scheme disagreement-guided token intervention (Figure 3). Experiments show that with a small intervention budget, this approach can substantially recover—and in some settings even surpass—the performance of a same-size reasoning model on challenging reasoning tasks. Together, these findings motivate a token-level view of reasoning: a small number of early planning commitments can steer the entire trajectory and dominate the reasoning gap.

## 2. Related Work

**Reasoning capabilities in base models.** Reasoning behaviors can emerge in small or base models when they are exposed to appropriate training signals, such as reasoning exemplars or Chain-of-Thought (Wei et al., 2022) traces (Ho et al., 2023). Explicit alignment, including fine-tuning on structured rationales, often suffices to elicit multi-step problem solving in models that previously answered directly (Ho et al., 2023). Reinforcement learning has been shown to gradually restructure output patterns, increasing response length and inducing more consistent use of intermediate reasoning steps (Guo et al., 2025; Wang et al., 2025b). Activation-space methods such as steering or targeted editing demonstrate that reasoning patterns, like backtracking, can be causally induced in base models without weight updates (Venhoff et al., 2025; Ward et al., 2025). Inference-time approaches, such as CoT prompting or self-verification feedback, can also trigger reasoning behaviors that do not naturally surface in zero-shot settings (Wei et al., 2022; Gandhi et al., 2025). Collectively, these findings suggest that base models possess a wide range of reasoning capabilities that can be activated through post-training mechanisms, guidance signals, or structural interventions.

**Token-level distributional disagreement.** Cross-entropy, entropy, and divergence offer effective signals for identifying decision-critical positions where models diverge or hesitate during generation. These signals can help locate forking tokens, error-prone steps, or places where weak models diverge sharply from stronger references. *KL divergence*, in both forward and reverse form, has been used to compare different models' behavior. *Forward KL* penalizes the student for deviating from the teacher's distribution, while *Reverse KL* emphasizes tokens where the student is overconfident in outputs disfavored by the teacher (Hinton et al., 2015; Gu et al., 2024; Agarwal et al., 2024). *Entropy* measures local uncertainty in the model's predictive distri-

bution. Entropy spikes often correspond to reasoning forks or hesitation. Focusing updates on high-entropy tokens improves RLVR efficiency (Wang et al., 2025a), while entropy-guided rollout branching improves exploration (Zheng et al., 2025). *Cross-entropy* measures the magnitude of divergence between a model's output and a target distribution, and is widely used as a training loss. In contrast, $\Delta \log p$, the difference between log probabilities in model before and after post-training, captures the directionality of model updates. Such metrics support the identification of high-impact tokens where reasoning behavior changes or supervision is most needed.

**Selective intervention at inference time.** The reasoning capabilities of weak models can be enhanced by inserting strong-model control without replacing the entire rollout. These approaches typically fall into two paradigms: collaborative inference for efficiency and guided generation for distillation stability. In collaborative inference, strong models serve as on-demand verifiers to balance performance and compute. At the request level, expensive resources can be reserved for hard queries by dispatching inputs based on predicted capability gaps (Ong et al., 2025). For finer-grained control, verification mechanisms allow strong models to screen and take over from lightweight drafters upon detecting quality degradation (Chen et al., 2023). Furthermore, difficult reasoning steps can be explicitly "relayed" to a strong model (Huang et al., 2026) or escalated to the cloud (Sun et al., 2024) specifically when small models exceed internal risk thresholds. Beyond inference acceleration, selective interventions are crucial for stabilizing on-policy distillation by preventing student models from learning from erroneous trajectories (Hinton et al., 2015; Gu et al., 2024; Agarwal et al., 2024). To ensure data quality, teacher distributions can serve as filters to reject unlikely tokens (Xu et al., 2025), or dynamically take over generation at divergence peaks to correct reasoning paths (Peng et al., 2025). Together, these strategies demonstrate that concentrating teacher assistance on sparse, high-impact positions effectively bridges the gap between weak and strong models in both inference and training stages.

# 3. Preliminary

In this paper, we focus on the token-level behavioral differences between a standard base model and a stronger reasoning model sharing the same vocabulary. In this section, we establish the framework for our analysis. We first introduce the comparative generation setup and the metrics for quantifying distributional disagreement in Section 3.1. Then, we formally formulate the problem of sparse reasoning takeover in Section 3.2.

## 3.1. Setup and Notations

**Autoregressive generation process.** To characterize the reasoning gap, we consider a standard autoregressive generation setting involving two models sharing a vocabulary $\mathcal{V}$: a base model $\mathcal{M}_b$ and a stronger reasoning model $\mathcal{M}_r$ (Vaswani et al., 2017; Brown et al., 2020; Hinton et al., 2015; Leviathan et al., 2023). Given a prompt $x_0$, the model generates a sequence $y = (y_1, \ldots, y_T)$. At each step $t$, let $x_t = (x_0, y_{<t})$ denote the context. We denote the next-token probability distributions as $p_b(\cdot \mid x_t)$ and $p_r(\cdot \mid x_t)$ respectively. Typically, $\mathcal{M}_r$ exhibits superior reasoning capabilities (e.g., via larger scale or specialized post-training) but incurs higher computational costs, whereas $\mathcal{M}_b$ is efficient but performance-limited (Kaplan et al., 2020; Ouyang et al., 2022; Wei et al., 2022). Our goal is to understand how $\mathcal{M}_b$ deviates from the superior trajectory of $\mathcal{M}_r$.

**Quantified disagreement metrics.** To quantify the behavioral disagreement between the two models at a granular level, we define a generic token-level disagreement score $s_t$ along the trajectory generated by the base model. Specifically, at step $t$ with context $x_t$, the disagreement is measured by a disagreement function $\mathcal{D}_f$ between the base model's distribution $p_b(\cdot \mid x_t)$ and the reasoning model's distribution $p_r(\cdot \mid x_t)$:

$$s_t = \mathcal{D}_f \big( p_b(\cdot \mid x_t), \, p_r(\cdot \mid x_t) \big) \qquad (1)$$

In this work, we primarily utilize the Cross Entropy (CE) as the disagreement metric (Cover & Thomas, 2001). It measures the expected surprisal of the reasoning model with respect to the base model's choices:

$$\begin{aligned} \mathcal{D}_f = \mathcal{D}_{\mathrm{CE}}(t) &= H(p_b(\cdot \mid x_t), p_r(\cdot \mid x_t)) \\ &= -\sum_{y \in \mathcal{V}} p_b(y \mid x_t) \log p_r(y \mid x_t). \end{aligned} \qquad (2)$$

We note that other metrics, such as the Reverse KL divergence, are also applicable and closely related (i.e., $D_{\mathrm{rKL}} = \mathcal{D}_{\mathrm{CE}} - H_b$). These metrics serve as the primary probing signals to localize where the base model deviates from the reasoning model's decision path. In parallel, we monitor the base model's intrinsic uncertainty using the Shannon entropy $H_b(t) = H(p_b(\cdot \mid x_t))$.

## 3.2. Problem Formulation

While the reasoning model $p_r$ outperforms the base model $p_b$, deploying it for every token generation is computationally prohibitive. Our objective is to bridge this performance gap by intervening only on a sparse subset of tokens. Formally, we aim to identify a set of *decision-critical positions* $\mathcal{S} \subset \{1, \ldots, T\}$, such that $|\mathcal{S}| \ll T$.

We define a disagreement-guided token intervention mechanism where the generation follows the base model $p_b$ by

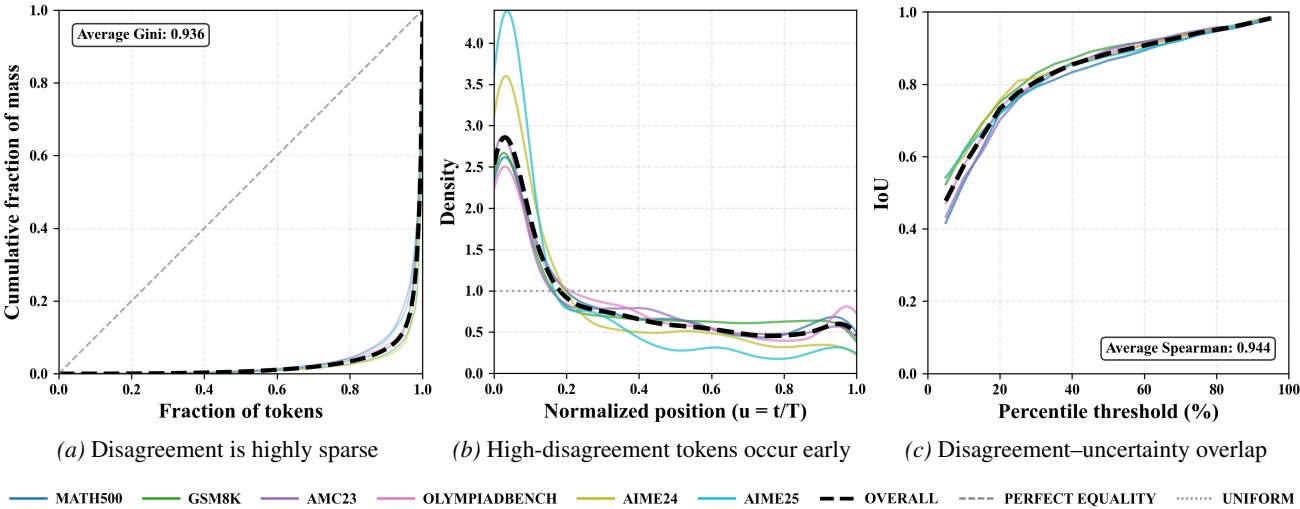

*Figure 2.* Token-level disagreement is sparse, early, and uncertainty-aligned across benchmarks. (a) Lorenz curves show that disagreement mass is highly concentrated in a small fraction of tokens. (b) Normalized-position density of the top-1% highest-disagreement tokens is strongly skewed toward early steps (u=t/T), indicating early-to-mid trajectory concentration relative to a uniform baseline. (c) IoU between top-$p\%$ disagreement tokens and top-$p\%$ entropy tokens remains high across thresholds, showing strong alignment between cross-model disagreement and base-model uncertainty.

default, but switches to the reasoning model $p_r$ at indices $t \in \mathcal{S}$. This can be viewed as a dynamic interplay where the base model handles execution tokens, and the reasoning model intervenes solely at decision tokens.

Our core hypothesis is that the performance gap is not uniformly distributed across all tokens but is concentrated at positions of high distributional disagreement. Therefore, the identification of $\mathcal{S}$ can be formulated as a detection problem using the disagreement metrics defined in Section 3.1. If $D_{\mathrm{CE}}(t)$ effectively captures these critical branching points, we can recover the reasoning capabilities of $\mathcal{M}_r$ with a minimal intervention budget. In the following sections, we will first analyze the distribution of $D_f$ to validate this sparsity hypothesis (Section 4), and then propose a concrete algorithm for budgeted intervention (Section 5).

## 4. Token-Level Base–LRM Disagreement

In this section, we provide a comprehensive analysis of the token-level distributional disagreement between base models and LRMs. Specifically, we have the following core findings: (i) token-level distributional disagreement happens at few token positions, which is sparse and heavy-tailed (*cf.* Section 4.1); (ii) token-level distributional disagreement tends to happen at early positions (*cf.* Section 4.2); (iii) token-level distributional disagreement tends to align with the uncertainty of the base model, concentrating on planning-related tokens (*cf.* Section 4.3); and (iv) token-level distributional disagreement degree can be utilized for predicting the final reasoning failures (*cf.* Section 4.4).

We conduct analysis mainly on the Qwen3 family (Yang

et al., 2025), employing the Qwen3-0.6B-Base model for rollout generation ($p_b$) and the Qwen3-8B (Thinking) for reasoning guidance ($p_r$). Evaluations are conducted across six mathematical benchmarks ranging from standard tasks ( GSM8K (Cobbe et al., 2021), MATH500 (Hendrycks et al., 2021)) to competition-level challenges ( AIME (Mathematical Association of America, 2024), OlympiadBench (He et al., 2024), AMC23 (AI-MO, 2023)). Detailed configurations for models, datasets, and baselines are provided in Appendix A. We follow the setup in Section 3.1 and use token-level disagreement $s_t$ (Cross Entropy by default) computed along the base rollout.

### 4.1. Disagreement Concentrates in Few Token Positions

> **Finding 1:** Disagreement mass is extremely concentrated: a tiny fraction of tokens accounts for most cross-model disagreement.

We study the per-token disagreement scores $\{s_t\}$ along base-model rollouts and ask whether the gap reflects diffuse, incremental drift or a small number of spikes. Figure 2a shows that token-level distributional disagreement is sparse and heavy-tailed, since the Lorenz curve lies far below the equality line (*i.e.,* most tokens accumulate little mass of the disagreement while a small subset dominates).

We summarize this concentration with the Gini coefficient, which quantifies how unevenly the cumulative disagreement mass is distributed across token positions (0 = uniform, 1 = maximally concentrated, *cf.* Appendix B.1) (Hurley & Rickard, 2009; Lorenz, 1905). Across six math reasoning

*Table 1.* **Planning enrichment at uncertainty/disagreement spikes.** We label tokens as *planning* vs. *execution* and report the share of planning tokens in the global token stream versus among top-1% tokens ranked by either base entropy $H_b(t)$ or cross-model divergence $s_t$. *Enrich.* is the ratio of the two shares.

| Metric | Global (%) | Top-$p$% (%) | Enrich. |
|---|---|---|---|
| Entropy $H_b(t)$ | 1.89 | 15.75 | 8.31$\times$ |
| Disagreement $s_t$ | 1.89 | 14.13 | 7.46$\times$ |

*Table 2.* Failure prediction performance. We compare failure prediction using base model uncertainty (Entropy) versus our cross-model disagreement (Cross Entropy), aggregated as the mean of top-100 tokens per sample.

| Dataset | Metric (Top-100 Mean) | AUROC $\uparrow$ | Acc $\uparrow$ |
|---|---|---|---|
| MATH500 | Entropy (Baseline) | 0.718 | 0.650 |
| | Disagreement (Ours) | 0.773 | 0.710 |
| GSM8K | Entropy (Baseline) | 0.817 | 0.760 |
| | Disagreement (Ours) | 0.851 | 0.840 |

benchmarks, we obtain an average $G = 0.936$ (details in Appendix B.1). This extreme inequality supports a sparse control view: the base–reasoning gap is driven primarily by rare, high-magnitude deviations at a small set of trajectory-sensitive positions, rather than by many small errors accumulating over time.

### 4.2. Disagreement Spikes Happen at Early Positions

> **Finding 2:** High-disagreement spikes are front-loaded: most occur in the earliest portion of generation rather than being spread uniformly across the rollout.

We examine the positional structure of disagreement by normalizing each token index as $u = t/T \in [0, 1]$ and selecting, for each sample, the top-1% tokens by disagreement score $s_t$. We then aggregate these selected positions across samples and estimate their density. As shown in Figure 2b, the resulting distribution is strongly left-skewed: spikes occur very early (mode at $u \approx 0.05$) and the density rapidly decays, becoming close to baseline by $u \approx 0.2$.

This pattern is inconsistent with a view where reasoning gains primarily come from long traces after the reasoning process unfolds. Instead, because early Chain-of-Thought tokens often encode decomposition and strategy selection (Wei et al., 2022), the positional skew supports a planning gap interpretation: the models diverge most at early trajectory-setting commitments, not during later execution.

### 4.3. Disagreement Spikes Relate to Model Uncertainty

> **Finding 3:** Disagreement spikes are not arbitrary: they coincide with base-model uncertainty spikes and are strongly enriched for planning-related tokens.

**Disagreement spikes coincide with intrinsic uncertainty.** We ask whether cross-model disagreement $s_t$ highlights structurally meaningful positions or merely reflects superficial stylistic differences. Figure 2c reports the Intersection-over-Union (IoU) between tokens ranked by disagreement score $s_t$ (CE-based) and those ranked by the base model entropy $H_b(t)$, computed per sample over the top-$p$% sets and averaged. The strong overlap shows that disagreement spikes concentrate at steps where the base model is intrinsi-

cally uncertain, providing an internal validation that these spikes correspond to difficult, trajectory-sensitive points rather than random mismatch.

**These uncertainty-linked disagreement spikes are planning-rich.** We further characterize the content of these positions. Using a lightweight heuristic classifier for *planning* vs. *execution* tokens (Appendix B.2), we compute planning enrichment among the top-1% tokens ranked by $s_t$ and $H_b(t)$ respectively within base-model rollouts (Subramanian et al., 2005). As shown in Table 1, planning markers are strongly over-represented: across all datasets, the planning share rises from 1.89% globally to 15.75% at entropy spikes (8.31$\times$) and to 14.13% at disagreement spikes (7.46$\times$). Appendix B.3 provides per-dataset results, confirming the trend is consistent across benchmarks.

Taken together, disagreement spikes systematically track uncertain planning commitments—positions where a small change in planning-related tokens can steer the subsequent trajectory.

### 4.4. Disagreement Spikes Predict Reasoning Failures

> **Finding 4:** Aggregating disagreement over a small set of high-disagreement tokens predicts reasoning failures.

The previous analyses suggest that disagreement is (i) extremely concentrated in rare spikes, (ii) front-loaded early in the rollout, and (iii) aligned with uncertain, planning-heavy steps. If these spikes indeed reflect trajectory-setting mistakes, then their magnitude within a sample should be predictive of whether the rollout ultimately succeeds.

**Failure score from disagreement spikes.** For each sample, we form a scalar score by averaging the top-$K$ ($K = 100$) per-token disagreement values $s_t$ within that sample. We evaluate how well this score predicts incorrect final answers using AUROC and Accuracy; we compute an analogous score using base entropy $H_b(t)$ for comparison.

As shown in Table 2 and Figure 4, the disagreement-based score provides a clear separation between correct and incorrect samples and consistently outperforms entropy (e.g.,

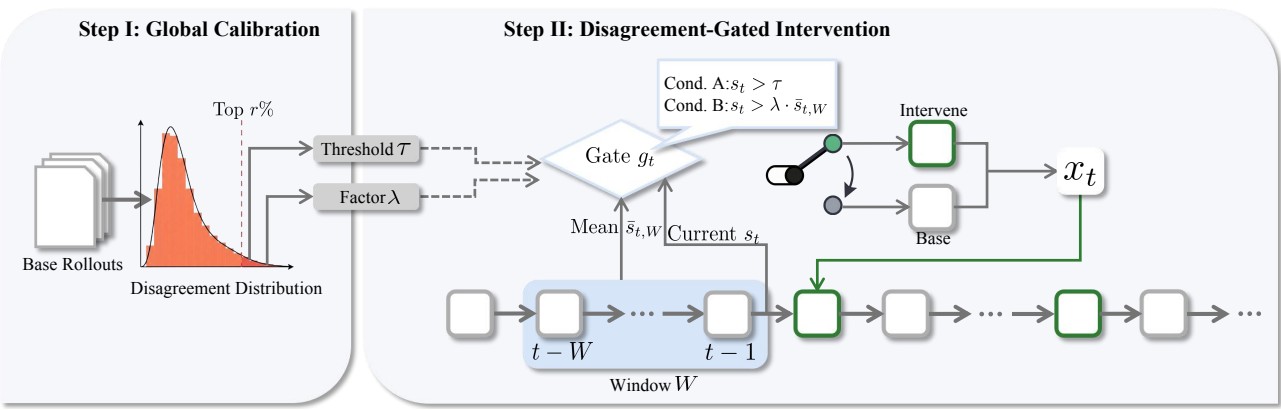

*Figure 3.* Illustration of Disagreement-Gated Intervention. In Step I, we calibrate the disagreement threshold $\tau$ and factor $\lambda$ using the top-$r\%$ of scores from base rollouts. In Step II, the gate $g_t$ dynamically switches between the Base model and the Intervene model (reasoning expert). Intervention is triggered only when the token-level disagreement $s_t$ satisfies both the global constraint ($s_t > \tau$) and the local relative spike condition ($s_t > \lambda \cdot \bar{s}_{t,W}$).

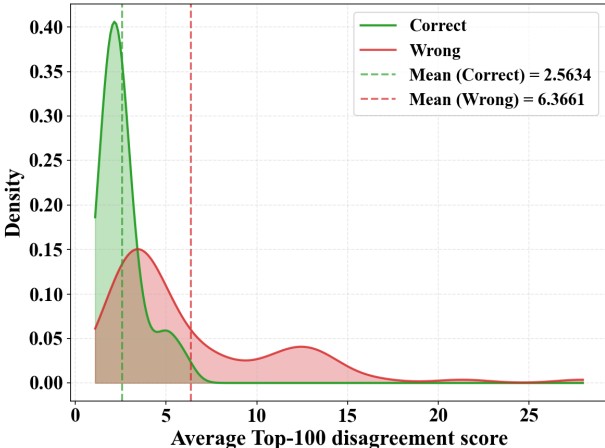

*Figure 4.* The distribution of top-100 mean disagreement scores shows a distinct separation, where incorrect samples (red) exhibit significantly higher values than correct ones (green).

GSM8K: 0.851 vs. 0.817 AUROC). This result closes the loop with the preceding structure findings: the same sparse, early, planning-linked spikes that dominate disagreement mass also carry actionable, sample-level predictive signal.

These observations motivate a question: if failure is driven by a small set of high-disagreement decisions, can selectively correcting only those positions recover strong-model reasoning under a small token budget? In the next section, we answer this question by introducing a disagreement-guided, token-level intervention at inference time.

## 5. Reasoning Can Be Restored by Correcting a Few Decision Tokens

Section 4 shows that base–reasoning disagreement is (i) extremely sparse and mostly happens at early positions of the trajectory, (ii) aligned with the base model's intrinsic uncertainty and enriched for planning markers, and (iii) predictive

of failure. Motivated by this sparse-control picture, we conduct an intervention test: can selectively correcting only a small set of disagreement spikes yield large performance recovery under a small replacement budget?

> **Finding 5:** A disagreement-gated, token-level replacement at a small set of spikes is sufficient to redirect rollouts and recover a disproportionate fraction of strong-model reasoning performance under small budgets.

### 5.1. Calibrated Disagreement Spikes for Gated Intervention

We design a minimal inference-time intervention that alters only a small subset of emitted tokens, leaving model parameters and hidden states unchanged. At each decoding step $t$, we compute a token-level disagreement score $s_t = \mathcal{D}_f(p_b(\cdot \,|\, x_t), p_r(\cdot \,|\, x_t))$, where $\mathcal{D}$ is a distributional disagreement between next-token distributions of the base and reasoning models. Unless otherwise stated, we instantiate $\mathcal{D}$ with cross entropy (CE), consistent with Section 4; we report results with reverse-KL as an alternative disagreement metric in Appendix C.1.

**Global calibration via a spike ratio $r$ (budget control).** To avoid per-task tuning, we calibrate a global spike criterion on a held-out calibration set by defining disagreement spikes as the top-$r$ tail of disagreement scores. Concretely, let $\mathcal{S} = \{s_t\}$ denote all disagreement scores collected along base rollouts on the calibration prompts. We set the global threshold $\tau$ as the $(1-r)$-quantile of $\mathcal{S}$:

$$\tau = Q_{1-r}(\mathcal{S}), \qquad (3)$$

so that $s_t > \tau$ corresponds to a top-$r$ disagreement event under the calibration distribution, and the calibrated thresholds transfer robustly across benchmarks (Appendix C.7).

*Table 3.* Experimental results on mathematical reasoning benchmarks (Accuracy / Pass@8). The Recovery column quantifies how much the intervention recovers reasoning capabilities, calculated as $(P_{Guided} - P_{Base})/(P_{Thinking} - P_{Base})$, where $P$ denotes Pass@8.

| Model | Math500 | GSM8K | AMC23 | Olympiad | AIME24 | AIME25 | Average | Recovery |
|---|---|---|---|---|---|---|---|---|
| Qwen3-0.6B-Base | 27.6 / 67.0 | 29.9 / 82.0 | 13.4 / 42.5 | 6.8 / 21.0 | 0.4 / 3.3 | 0.0 / 0.0 | 13.0 / 36.0 | - |
| + Guided ($\bar{\rho} \approx 0.04$) | 44.6 / 89.0 | 56.5 / 96.0 | 35.9 / 77.5 | 20.2 / 49.0 | 10.0 / 36.7 | 7.5 / 20.0 | 29.1 / 61.4 | 91% |
| + Guided ($\bar{\rho} \approx 0.13$) | 73.9 / 99.0 | 80.4 / 99.0 | 58.4 / 90.0 | 42.5 / 65.0 | 32.5 / 70.0 | 26.7 / 56.7 | 52.4 / 80.0 | 157% |
| Qwen3-0.6B (Thinking) | 72.1 / 90.0 | 81.6 / 94.0 | 46.9 / 80.0 | 34.3 / 54.0 | 9.2 / 30.0 | 16.1 / 36.7 | 43.4 / 64.1 | - |
| Qwen3-1.7B-Base | 54.2 / 82.0 | 75.4 / 93.0 | 28.1 / 65.0 | 19.2 / 41.0 | 5.0 / 20.0 | 2.5 / 10.0 | 30.8 / 51.8 | - |
| + Guided ($\bar{\rho} \approx 0.06$) | 66.8 / 95.0 | 72.0 / 97.0 | 50.6 / 95.0 | 33.2 / 64.0 | 20.4 / 46.7 | 18.8 / 43.3 | 43.6 / 73.5 | 76% |
| + Guided ($\bar{\rho} \approx 0.16$) | 81.5 / 100.0 | 88.9 / 98.0 | 71.9 / 97.5 | 48.9 / 67.0 | 45.0 / 73.3 | 36.2 / 66.7 | 62.1 / 83.8 | 112% |
| Qwen3-1.7B (Thinking) | 88.9 / 100.0 | 90.9 / 97.0 | 80.0 / 95.0 | 52.5 / 63.0 | 39.2 / 73.3 | 32.8 / 53.3 | 64.1 / 80.3 | - |
| Qwen3-8B (Thinking) | 97.4 / 100.0 | 97.8 / 98.0 | 90.9 / 100.0 | 60.0 / 69.0 | 70.0 / 83.3 | 52.5 / 73.3 | 78.1 / 87.3 | - |

We additionally compute a tail-to-mean scale factor

$$\lambda = \frac{\mathbb{E}[s \mid s > \tau]}{\mathbb{E}[s]}, \qquad (4)$$

which summarizes how much larger disagreement scores are within the top-$r$ tail compared to the global average.

**Runtime gating.** During decoding, we trigger takeover only when $s_t$ forms a robust spike under a two-part gate:

$$g_t = \mathbb{I}\Big[s_t > \tau \wedge s_t > \lambda \cdot \bar{s}_{t,W}\Big], \quad \bar{s}_{t,W} = \frac{1}{W} \sum_{i=1}^{W} s_{t-i}, \quad (5)$$

where the global threshold $s_t > \tau$ enforces a calibrated tail event, and the sliding-window ratio test suppresses noisy toggling by requiring the current score to exceed the recent local baseline by a factor $\lambda$. Token generation then follows

$$y_t \sim \begin{cases} p_r(\cdot \mid x_t), & g_t = 1, \\ p_b(\cdot \mid x_t), & g_t = 0. \end{cases} \qquad (6)$$

We also report the realized intervention rate $\rho$, the fraction of tokens generated by the reasoning model: $\rho = \frac{1}{T} \sum_{t=1}^{T} g_t$. The per-problem realized rate $\rho_i$ is concentrated with small variance across problems (Appendix C.2).

**Remark on practical deployment.** The disagreement gate queries both $p_b$ and $p_r$ at every step and is best viewed as a diagnostic form of sparse control rather than an optimized inference system. A cheaper alternative gates on the base entropy $H_b(t)$ alone, which already recovers most of the strong-model Pass@8 at small budgets (Appendix C.3).

## 5.2. Sparse Interventions Yield Disproportionate Performance Recovery

We evaluate how much reasoning performance can be restored by intervening only at disagreement spikes. Because a fixed spike ratio $r$ induces different takeover frequencies across prompts, we report results against the realized intervention rate $\rho = \frac{1}{T} \sum_{t=1}^{T} g_t$.

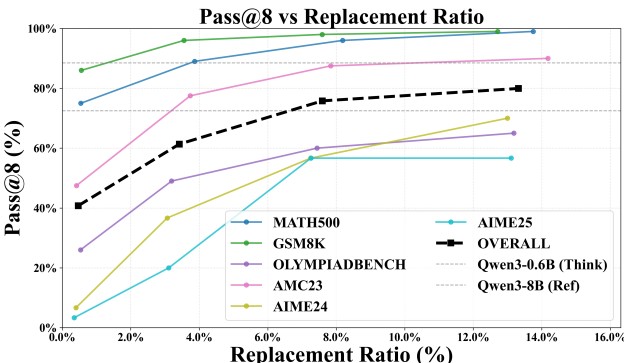

*Figure 5.* Performance recovers sharply with few replacement.

**A small budget can outperform a same-size reasoning model.** Table 3 shows that sparse takeover is highly effective. For Qwen3-0.6B, guided decoding at $\bar{\rho} \approx 0.13$ improves the average from $13.0/36.0$ to $52.4/80.0$ (Acc./Pass@8), surpassing the same-size thinking variant ($43.4/64.1$) and recovering $157\%$ of its Pass@8 gap. Qwen3-1.7B exhibits the same behavior: at $\bar{\rho} \approx 0.16$, guided decoding reaches $62.1/83.8$, essentially matching its thinking counterpart ($64.1/80.3$; $112\%$ recovery). A sample-level analysis further confirms that the intervention is rarely harmful: across 400 problems, it flips 152 examples from incorrect to correct and only 3 in the reverse direction (Appendix C.8).

**Recovery is sharply nonlinear in replacement ratio.** Figure 5 reveals a pronounced knee: most of the recovery arrives within the first few percent replacement, followed by clear diminishing returns. Concretely, the overall Pass@8 jumps by roughly $\sim 20$ points within the first few percent replacement (from $\approx 41\%$ at $\rho \approx 0.5\%$ to $\approx 61\%$ at $\rho \approx 3\%$), while later increases in $\rho$ deliver progressively smaller gains (saturating near $\approx 80\%$ by $\rho \approx 13\%$). Notably, around $\rho \approx 7\% - 8\%$ the overall curve already reaches the Qwen3-0.6B (Thinking) reference line, and approaching the 8B reference with larger budgets.

**Position selection, not strong-model injection, drives the**

```
\ boxed { 1 0 }
This is the solution :
Team B scores  2 goals less than team A , which is  2 +  4 =  6
goals . In the second half , team A scores  1 / 4 of the goals
scored by team B , which is  4 *  6 =  2 4 goals . Team B scores  4
times the number of goals it scored in the first half , which is
4 *  2 =  8 goals . The total number of goals scored in the match
is  4 +  6 +  2 4 +  8 = \ boxed { 4 2 } goals .
This is the solution :
Team A scores  4 goals in the first half , team B scores  4 -  2 =
2 goals . In the second half , team A scores  1 / 4 of team B 's
goals , which is  2 *  1 / 4 =  1 / 2 . Team B scores  4 times the
number of goals it scored in the first half , which is  4 *  2 =  8
goals . The total number of goals scored in the match is  4 +  2 +
1 / 2 +  8 = \ boxed { 1 3 . 5 } goals . This is the solution :
Team A scores  4 goals in the first half , team B scores  4 -  2 =
2 goals . In the second half , team A scores  1 / 4 of team B 's
goals , which is  2 *  1 / 4 =  1 / 2 . Team B scores  4 times the
number of goals it scored in the first half , which is  4 *  2 =  8
goals . The total number of goals scored in the match is  4 +  2 +
1 / 2 +  8 = \ boxed { 1 3 . 5 } goals .
[... Model enters a repetition loop and continues
indefinitely...]
```

```
Okay , let 's try to figure out the total number
of goals scored in the match . Hmm , let 's break
it down step by step . [ ... ]
2 goals fewer than team A . So , team B scored 4
– 2 = 2 goals in the first half . Wait , that
seems straightforward .
Now , moving to the second half . The problem states
that team A scores [ ... ]
the second half . But here , it says that team B
scores 4 times the number of goals it scored in
the first half .
Wait , that seems important . Let me double –check
that .
Team B 's goals in the first half [ ... ]
( first half ) + 8 ( second half ) = 1 0 goals .
Therefore , the total number of goals scored in the
match is 6 + 1 0 = 1 6 goals .
Wait , let me check again to [ ... ]
### Final Answer
$$
\ boxed { 1 6 }
$$
```

*(a)* Base: brittle parsing and degeneration          *(b)* Guided: brief stop-and-check resolves the ambiguity

*Figure 6.* **Qualitative pattern: takeover induces a "stop-and-check" planning step.** Green highlights tokens generated under takeover (reasoning model); white tokens are generated by the base model. In this example, a short self-check triggered at an ambiguous clause fixes the variable binding, after which control returns to the base model for routine execution.

*Table 4.* **Matched-budget baselines (Qwen3-0.6B).** Average Acc / Pass@8 over six math benchmarks. Guided intervention at $\bar{\rho} \approx$ 0.04 exceeds both baselines at $\rho = 0.25$.

| Method | Budget $\rho$ | Avg Acc | Avg Pass@8 |
|---|---|---|---|
| Base | 0.00 | 13.0 | 36.0 |
| + Random | 0.25 | 26.4 | 55.2 |
| + Early-only | 0.25 | 25.7 | 58.4 |
| + Guided (Ours) | $\bar{\rho} \approx 0.04$ | **29.1** | **61.4** |

*Table 5.* **Enrichment Analysis.** We compare the token distribution in the global generation vs. our intervention set. Planning tokens are effectively targeted.

| Category | Global (%) | Interv. (%) | Enrich. |
|---|---|---|---|
| **Planning** | 1.9 | 33.3 | **17.6×** |
| Execution | 98.1 | 66.7 | 0.7× |

magnitude and preserve the recovery curve (Appendix C.7).

Taken together, these results show that the base–reasoning gap can be closed largely by correcting a sparse set of high-disagreement decisions: a few-percent replacement already matches or exceeds the size-matched thinking model.

### 5.3. Intervention Induces Reasoning Patterns

Beyond final accuracy, we ask what the gate intervenes on and how those interventions change the trajectory. We find a consistent functional pattern: takeover is concentrated on planning markers and often manifests as a brief stop-and-check step that resolves an upstream ambiguity before handing control back to the base model.

**Takeover targets planning rather than execution.** Table 5 quantifies the token types in our takeover set. Although planning tokens constitute only 1.9% of classifiable tokens in base rollouts, they account for 33.3% of takeover tokens (17.6× enrichment). Conversely, execution tokens are under-represented (0.7×), suggesting that the gate largely trusts the base model for routine arithmetic once the high-level structure is fixed.

**gain.** A natural concern is whether the recovery merely reflects that any injection of strong-model tokens helps. We rule this out with two matched-budget baselines: a random baseline that uniformly samples replacement positions, and an early-only baseline that replaces the first $\rho \cdot L_r$ tokens (motivated by Finding 2's early skew). As shown in Table 4, both baselines at an aggressive $\rho = 0.25$ are dominated by guided intervention at only $\bar{\rho} \approx 0.04$—a ∼6× smaller budget. This shows that the gain stems from which tokens are corrected rather than from invoking the strong model alone, and that position alone is not sufficient: within the early region, only the disagreement spikes are truly trajectory-setting. Full per-budget curves are provided in Appendix C.4.

**Generalization beyond Qwen3.** The same sparse-control behavior transfers to a LLaMA pair (LLaMA-3.1-8B as base, DeepSeek-R1-Distill-Llama-8B as guide), where intervening on ∼20% of tokens recovers ∼91% of the Pass@8 gap (Appendix C.5). The calibration is also robust: thresholds re-fitted on each benchmark stay within the same order of

**A representative example: induced stop-and-check planning.** Figure 6 illustrates how this specialization appears in generation. The base model fails to stably bind variables under a syntactically ambiguous clause and degenerates. Under takeover, the reasoning model injects a short self-check that disambiguates the dependency, after which decoding returns to the base model for straightforward computation. We provide the full prompt and additional qualitative examples in Appendix D. This planning-specialized takeover provides a mechanistic account for the sharp budget–performance curve in Figure 5: a small number of corrected planning commitments can steer long downstream execution.

## 6. Limitations

This work has several limitations. First, due to compute constraints, our study primarily focuses on the Qwen3 family and math-centric reasoning benchmarks; we provide initial evidence that the same sparse-control phenomenon transfers to a LLaMA-based pair (Appendix C.5) and to non-math scientific QA on GPQA-Diamond (Appendix C.6), but broader coverage of model families, model sizes, and reasoning domains (e.g., code, multi-hop QA, agentic reasoning) remains future work. Second, as a diagnostic-first findings paper, we adopt a simple, transparent intervention rule and do not aim to exhaustively optimize system efficiency or compete with heavily engineered baselines.

## 7. Conclusion

We investigated the base–reasoning gap through token-level distributional disagreement. Across reasoning benchmarks, disagreement highly concentrates in rare spikes that cluster early, align with base-model uncertainty, and are enriched for planning-related decision tokens; aggregating these spikes further predicts eventual failures. Motivated by these findings, we introduced a globally calibrated, disagreement-gated token intervention that edits only a small subset of tokens. Under modest realized takeover budgets, this intervention recovers a disproportionate fraction of strong-model reasoning performance and can outperform a same-size reasoning variant. Qualitatively, intervention primarily targets planning rather than execution. Mechanistically, because generation is autoregressive, correcting a few early planning-related tokens redirects the subsequent trajectory while leaving routine execution to the base model. Overall, our results support a sparse control view of reasoning: a few early planning commitments can steer long downstream trajectories, enabling selective upgrades only at branching points.

## Impact Statement

This paper studies the token-level distributional disagreement between base models and large reasoning models (LRMs), and shows that token-level interventions can help the base model largely recover the LRM's reasoning capability. Generally, this paper may yield a positive societal impact by helping us understand the underlying mechanism of LLM reasoning.

## Acknowledgements

This work was partially supported by the USTC Kunpeng-Ascend Scientific and Education Innovation Excellence Center.

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

# A. Experimental Setup

## A.1. Implementation Details

**Compute Environment and Models.** We implement all experiments in PyTorch[1] using the Hugging Face Transformers library, and rely on vLLM (Kwon et al., 2023) for efficient, high-throughput inference. All evaluations are conducted on a single compute node equipped with an NVIDIA A100 GPU. To enable a controlled comparison between base generation and reasoning guidance (with a shared tokenizer and vocabulary), we use the Qwen3 model family (Yang et al., 2025). Specifically, we use `Qwen3-0.6B-Base` as the primary rollout generator ($\mathcal{M}_b$) for its efficiency, and `Qwen3-8B` as the stronger reference model ($\mathcal{M}_r$), which provides disagreement scores and takeover tokens when interventions are triggered.

**Prompting and Inference Configuration.** We wrap each problem using the following template and employ sampling-based decoding with temperature $0.6$ and top-$p$ $0.95$. Unless otherwise specified, we generate $K = 8$ samples per problem in parallel. To accommodate long chain-of-thought traces in competition-level mathematics, we set the maximum generation length to 16,384 tokens.

> **PROMPT:**
>
> Please reason step by step, and put your final answer within \boxed{}.
> This is the problem:
> {{prompt}}

**Sampling Protocol and Metrics.** For each problem, we draw $K = 8$ independent samples under the same decoding configuration. We report *Pass@8* as whether any of the 8 samples is verified as correct. For *Accuracy*, we report the average correctness across the same 8 samples (i.e., the mean of 8 Bernoulli outcomes), rather than greedy decoding.

**Answer Verification.** We verify predicted answers using the `math_verify` library. Concretely, we parse both the gold answer and the model response using a joint extraction configuration (`ExprExtractionConfig` and `LatexExtractionConfig`), and then apply `verify(gold, pred)` to test mathematical equivalence. If parsing fails or the verifier raises an exception (e.g., due to unsupported SymPy objects), we conservatively mark the prediction as incorrect.

**Intervention Mechanism and Calibration.** Our disagreement-gated intervention is grounded in Cross Entropy (CE) between the next-token distributions of the base and reference models. Intervention thresholds are calibrated on a held-out set of 500 prompts randomly sampled from the GSM8K training split (Cobbe et al., 2021), ensuring no overlap with any test set. As reported in our results, we evaluate two primary sparsity settings, $r = 0.03$ and $r = 0.05$, targeting the top 3% and 5% of disagreement spikes, respectively. To improve stability and reduce spurious triggers, we apply sliding-window smoothing (see Equation (5)) with window size $W = 64$, intervening only when the current disagreement significantly exceeds the local moving average.

**Pseudocode.** Algorithm 1 provides a self-contained pseudocode for the disagreement-gated intervention. The calibration step (lines 1–4) is run once on a held-out set; the runtime step (lines 5–14) is invoked online during decoding. The procedure does not modify any model parameters or hidden states.

---

[1] https://pytorch.org/

---

**Algorithm 1** Disagreement-Guided Token Intervention

---

**Input:** base model $p_b$, reasoning model $p_r$, calibration prompts $\mathcal{C}$, target spike ratio $r$, sliding-window size $W$, max length $T_{\max}$, prompt $x_0$.

**Output:** response $y_{1:t}$.

1: // — *Phase I: Offline calibration (run once)* —
2: Run base rollouts on $\mathcal{C}$ and collect $\mathcal{S} = \{s_t\}$ with $s_t = \mathrm{CE}(p_b(\cdot\,|\,x_t),\ p_r(\cdot\,|\,x_t))$
3: $\tau \leftarrow Q_{1-r}(\mathcal{S})$                                                     // $(1-r)$-*quantile threshold*
4: $\lambda \leftarrow \mathbb{E}[s \mid s > \tau]/\mathbb{E}[s]$                             // *tail-to-mean scale*
5: // — *Phase II: Online decoding for prompt* $x_0$ —
6: Initialize $y_{1:0} \leftarrow \emptyset$
7: **for** $t = 1, \ldots, T_{\max}$ **do**
8: $\quad x_t \leftarrow (x_0, y_{1:t-1})$
9: $\quad$ Compute $s_t = \mathrm{CE}(p_b(\cdot\,|\,x_t),\ p_r(\cdot\,|\,x_t))$
10: $\quad \bar{s}_{t,W} \leftarrow \frac{1}{\min(t-1,\,W)} \sum_{i=1}^{\min(t-1,\,W)} s_{t-i}$    // *skip gate if* $t = 1$
11: $\quad g_t \leftarrow \mathbb{I}\big[s_t > \tau \ \wedge\ s_t > \lambda \cdot \bar{s}_{t,W}\big]$
12: $\quad$ **if** $g_t = 1$ **then**
13: $\quad\quad y_t \sim p_r(\cdot\,|\,x_t)$                                                   // *one-token takeover*
14: $\quad$ **else**
15: $\quad\quad y_t \sim p_b(\cdot\,|\,x_t)$
16: $\quad$ **end if**
17: $\quad$ **if** $y_t = \texttt{<EOS>}$ **then**
18: $\quad\quad$ **break**
19: $\quad$ **end if**
20: **end for**
21: **return** $y_{1:t}$

---

### A.2. Evaluation Benchmarks

We evaluate the proposed method and baselines on a comprehensive suite of mathematical reasoning benchmarks, categorized into standard tasks (GSM8K and MATH500) and competition-level challenges (AMC23, AIME24, AIME25, and OlympiadBench), along with a non-math scientific QA benchmark (GPQA-Diamond) to assess cross-domain generalization.

- **GSM8K** (Grade School Math 8K) (Cobbe et al., 2021): A dataset consisting of 8.5K high-quality grade school math word problems. It requires models to perform multi-step elementary reasoning using basic arithmetic operations. We use the standard test set to evaluate the model's ability to handle linguistic diversity and sequential logic in relatively simple scenarios.

- **MATH500** (Hendrycks et al., 2021): A representative subset of the challenging MATH dataset, containing 500 problems sampled across seven categories (e.g., algebra, geometry, number theory) and five difficulty levels. MATH500 provides a computationally efficient yet rigorous estimate of a model's performance on broader pre-university competition mathematics.

- **AMC23** (AI-MO, 2023): Comprises problems from the 2023 American Mathematics Competitions (AMC 10 and AMC 12). These exams are the first tier in the United States high school mathematics competition series, requiring creative problem-solving skills beyond standard curriculum application.

- **AIME24 & 25** (Mathematical Association of America, 2024): The American Invitational Mathematics Examination is the second tier of the competition series, accessible only to top performers in the AMC. The problems in this set involve integers between 0 and 999 as answers and require significantly deeper reasoning and sustained logical chains compared to the AMC level.

- **OlympiadBench** (He et al., 2024): A comprehensive bilingual (English and Chinese) benchmark featuring Olympiad-level mathematics and physics problems. We focus on the mathematics subset, which sources problems from prestigious competitions such as the International Mathematical Olympiad (IMO) and the Chinese College Entrance Examination (GaoKao). This represents the upper bound of difficulty in our evaluation suite.

- **GPQA-Diamond** (Rein et al., 2024): A benchmark of graduate-level scientific question answering across biology, physics, and chemistry, written and validated by domain experts. The Diamond subset contains the most difficult "Google-proof" questions, requiring deep domain knowledge rather than retrieval. We use a 50-question subset to evaluate generalization beyond mathematical reasoning, with results reported in Appendix C.6.

# B. Analysis Details

## B.1. Lorenz Curve and Gini Coefficient for Token-Level Discrepancy

This section provides the formal definitions of the Lorenz curve and the Gini coefficient used in Section 4.1, as well as implementation details.

**Token-level disagreement scores.** For each response, we obtain a sequence of nonnegative per-token scores $\{s_t\}_{t=1}^N$ (e.g., cross-entropy discrepancy between base and reasoning next-token distributions). We sort these scores in non-decreasing order: $s_{(1)} \leq s_{(2)} \leq \cdots \leq s_{(N)}$.

**Lorenz curve.** (Lorenz, 1905) The Lorenz curve is defined by points $(x_k, y_k)$ for $k = 1, \ldots, N$, where

$$x_k = \frac{k}{N}, \qquad y_k = \frac{\sum_{i=1}^k s_{(i)}}{\sum_{j=1}^N s_{(j)}}. \tag{7}$$

Intuitively, $y_k$ measures what fraction of total discrepancy mass is contributed by the lowest-scoring $k$ tokens. A curve close to the equality line $y = x$ indicates a nearly uniform contribution across tokens, while a strongly bowed curve indicates high concentration.

**Gini coefficient.** (Hurley & Rickard, 2009) We summarize concentration by the Gini coefficient $G$, which equals twice the area between the Lorenz curve and the equality line:

$$G = 1 - 2 \int_0^1 L(x)\, dx. \tag{8}$$

For a discrete set of sorted scores, we compute

$$G = \frac{\sum_{i=1}^N (2i - N - 1)\, s_{(i)}}{N \sum_{j=1}^N s_{(j)}}, \tag{9}$$

which ranges from 0 (perfectly uniform mass) to 1 (all mass on a single token).

**Aggregation across samples and datasets.** We compute the Lorenz curve and Gini coefficient per sample using that sample's token sequence, and then report the dataset-level mean (and optionally standard deviation / confidence intervals) across samples. The overall value reported in Section 4.1 is the average of dataset-level means across the six benchmarks.

## B.2. Token Enrichment Analysis: Definition and Token Categorization

Section 4.3 and Section 5.3 report two complementary token enrichment analyses (Subramanian et al., 2005) using the same token categorization rules. The first ( peak-based enrichment) is computed on base-model rollouts, measuring whether tokens at high-entropy or high cross-model discrepancy peaks are planning-heavy. The second ( takeover-based enrichment) is computed on guided rollouts, measuring whether the tokens actually generated by the reasoning model during takeover are enriched for planning markers. This appendix provides the shared enrichment definition and the token categorization procedure used in both analyses.

**Enrichment ratio.** Let $C$ be a token category (e.g., *Planning*). We compute an enrichment ratio

$$E(C) \;=\; \frac{P(t \in C \mid t \in \mathcal{T}_{\text{interv}}^{\text{sem}})}{P(t \in C \mid t \in \mathcal{T}_{\text{global}}^{\text{sem}})}, \tag{10}$$

where $\mathcal{T}_{\text{interv}}^{\text{sem}}$ denotes the set of intervened tokens (i.e., tokens generated by the reasoning model during takeover) that are classifiable into one of our semantic categories, and $\mathcal{T}_{\text{global}}^{\text{sem}}$ denotes the set of classifiable tokens in the standard base rollout. Importantly, tokens that do not match any semantic rule (e.g., punctuation, stopwords, generic content words) are excluded from both the numerator/denominator counts and the normalizing totals, so the enrichment is computed within the planning/execution semantic subset.

**Token categorization (heuristic).** We classify each decoded token $t$ into one of two categories: (i) *Planning/Decision* and (ii) *Execution/Arithmetic*. Classification uses a lightweight heuristic based on exact keyword matching (for planning) and regular-expression/symbol matching (for execution). All comparisons are case-insensitive after stripping surrounding whitespace.

**Planning / Decision Tokens.** This category includes words associated with reasoning structure, strategy selection, logical deduction, and self-reflection. We constructed a vocabulary $\mathcal{V}_{\text{plan}}$ containing several keywords across the following sub-categories:

- **Step Indicators:** `first`, `next`, `then`, `finally`, `step`.

- **Logical Connectives:** `therefore`, `so`, `since`, `because`, `thus`, `hence`, `implies`.

- **Strategy & Action Verbs:** `let`, `assume`, `suppose`, `consider`, `calculate`, `solve`, `check`, `verify`, `notice`, `determine`, `choose`, `analyze`.

- **Contrast & Correction:** `but`, `however`, `actually`, `wait`, `instead`, `alternatively`.

- **Thinking Markers:** `thinking`, `strategy`, `idea`, `plan`, `break down`, `approach`, `method`.

- **Conditionals & Wh-Words:** `if`, `unless`, `whether`, `why`, `how`, `what`.

**Execution / Arithmetic Tokens.** This category captures the mechanical execution of mathematical operations and symbolic manipulation. We identified these tokens using regular expressions:

- **Numbers:** Integers and floating-point numbers (e.g., `100`, `3.14`).

- **Operators:** Mathematical symbols including $+$, $-$, $*$, $/$, $=$, $<$, $>$, `^`.

- **Structure:** LaTeX formatting symbols and brackets (e.g., `$`, `{`, `}`, `(`, `)`).

- **Variables:** Single-letter variables common in math problems (e.g., `x`, `y`, `n`) when appearing in a mathematical context.

**Target vs. reference token sets.** Each enrichment analysis compares a target token set $\mathcal{T}_{\text{target}}$ to a reference token set $\mathcal{T}_{\text{ref}}$, and computes category proportions within the classifiable semantic subset. In the peak-based enrichment (Section 4.3), $\mathcal{T}_{\text{target}}$ is the top-$p\%$ tokens in a base rollout ranked by either base entropy $H_b(t)$ or cross-model discrepancy $s_t$, and $\mathcal{T}_{\text{ref}}$ is the full base-rollout token stream. In the takeover-based enrichment (Section 5.3), $\mathcal{T}_{\text{target}}$ consists of takeover tokens extracted from guided rollouts using the per-step provenance field `chosen_from` (sources beginning with `teacher`), and $\mathcal{T}_{\text{ref}}$ is the corresponding base-rollout token stream.

**Interpretation and limitations.** This enrichment analysis is intended as a coarse, interpretable proxy for identifying planning markers. While it does not capture all planning behavior (e.g., implicit planning without explicit markers), the strong enrichment of planning tokens in the intervention set supports the claim that our takeover mechanism disproportionately targets planning-related linguistic cues rather than execution-heavy arithmetic tokens.

### B.3. Per-dataset planning enrichment at entropy/disagreement peaks

Table 1 aggregates planning enrichment across all datasets. Here we provide the per-dataset breakdown to assess consistency and heterogeneity across benchmarks. Table 6 shows that planning enrichment at top-1% peaks holds across all datasets for both ranking signals, with particularly large effects on harder benchmarks (e.g., AMC23 and AIME25).

*Table 6.* **Planning enrichment at peaks across datasets (top-1%).** For each dataset, we report the planning share in the base-rollout global token stream and among the top-1% tokens selected by either cross-model divergence $s_t$ (CE) or base entropy $H_b(t)$, together with the enrichment ratio.

| | Disagreement $s_t$ (CE) | | | Entropy $H_b(t)$ | | |
|---|---|---|---|---|---|---|
| Dataset | Global (%) | Top-1% (%) | Enrich. | Global (%) | Top-1% (%) | Enrich. |
| Math500 | 2.22 | 15.98 | 7.21× | 2.22 | 15.90 | 7.17× |
| GSM8K | 1.42 | 5.19 | 3.65× | 1.42 | 7.39 | 5.19× |
| AMC23 | 1.63 | 16.37 | 10.05× | 1.63 | 17.94 | 11.01× |
| OlympiadBench | 1.92 | 14.19 | 7.41× | 1.92 | 14.53 | 7.59× |
| AIME24 | 1.96 | 15.41 | 7.86× | 1.96 | 13.28 | 6.77× |
| AIME25 | 1.72 | 29.18 | 16.98× | 1.72 | 34.95 | 20.34× |

## C. Additional Results

### C.1. Reverse-KL as an Alternative Disagreement Signal

Our method is not tied to Cross Entropy as the disagreement metric. We replicate the gated takeover using reverse KL, i.e., $s_t = \text{KL}(p_b \| p_r)$, while keeping the same calibration protocol (global spike ratio $r$ and sliding-window smoothing with $W = 64$) and the same evaluation setup. Table 7 reports the results and shows that reverse-KL yields strong recovery under comparable realized budgets, confirming that divergence-guided takeover is not specific to a single discrepancy function.

*Table 7.* Experimental results on mathematical reasoning benchmarks (Accuracy / Pass@8). The Recovery column quantifies how much the intervention recovers reasoning capabilities, calculated as $(P_{Guided} - P_{Base})/(P_{Thinking} - P_{Base})$, where $P$ denotes Pass@8 (consistent with Table 3).

| Model | Math500 | GSM8K | AMC23 | Olympiad | AIME24 | AIME25 | Average | Recovery |
|---|---|---|---|---|---|---|---|---|
| Qwen3-0.6B-Base | 27.6 / 67.0 | 29.9 / 82.0 | 13.4 / 42.5 | 6.8 / 21.0 | 0.4 / 3.3 | 0.0 / 0.0 | 13.0 / 36.0 | - |
| + Guided ($\bar{\rho} \approx 0.08$) | 82.9 / 99.0 | 87.4 / 98.0 | 64.4 / 92.5 | 42.4 / 58.0 | 27.9 / 53.3 | 27.5 / 46.7 | 55.4 / 74.6 | 137% |
| + Guided ($\bar{\rho} \approx 0.17$) | 90.6 / 100.0 | 93.8 / 99.0 | 84.1 / 97.5 | 47.9 / 62.0 | 40.8 / 60.0 | 35.0 / 60.0 | 65.4 / 79.8 | 156% |
| Qwen3-0.6B (Thinking) | 72.1 / 90.0 | 81.6 / 94.0 | 46.9 / 80.0 | 34.3 / 54.0 | 9.2 / 30.0 | 16.1 / 36.7 | 43.4 / 64.1 | - |
| Qwen3-8B (Thinking) | 97.4 / 100.0 | 97.8 / 98.0 | 90.9 / 100.0 | 60.0 / 69.0 | 70.0 / 83.3 | 52.5 / 73.3 | 78.1 / 87.3 | - |

*Table 8.* **Recovery under different disagreement metrics (Qwen3-0.6B).** We report Recovery computed on Accuracy and Pass@8 respectively.

| Metric | Budget | Recovery (Acc) | Recovery (Pass@8) |
|---|---|---|---|
| CE | $\bar{\rho} \approx 0.04$ | 53% | 91% |
|  | $\bar{\rho} \approx 0.13$ | 130% | 157% |
| rKL | $\bar{\rho} \approx 0.08$ | 140% | 137% |
|  | $\bar{\rho} \approx 0.17$ | 172% | 156% |

To better understand metric-dependent behavior, Table 8 summarizes Recovery computed on Accuracy and on Pass@8, respectively. Empirically, reverse-KL tends to yield higher Accuracy-based recovery, whereas Cross Entropy yields stronger Pass@8-based recovery. A plausible explanation is that $\text{KL}(p_b \| p_r)$ emphasizes cases where the base model places overly concentrated mass on an incorrect token (confidence miscalibration), while Cross Entropy more directly reflects distributional mismatch and can better improve the probability that at least one sampled trajectory reaches the correct solution.

### C.2. Stability of the Realized Replacement Budget

We examine the per-problem realized replacement rate $\rho_i = \frac{1}{T_i} \sum_{t=1}^{T_i} g_t$. Figure 7 visualizes the distribution of $\rho_i$ across benchmarks for Qwen3-0.6B under Cross-Entropy and reverse-KL gating, each at two calibrated spike ratios ($r = 0.03$ and $r = 0.05$).

Two observations are consistent across datasets. First, the distributions are concentrated with small variance, indicating that the realized intervention budget is stable across problems rather than dominated by a small subset of outliers. Second, for the same $r$, Cross Entropy consistently yields a lower realized replacement rate than reverse-KL, suggesting that CE produces a more selective gate under our calibration procedure. This motivates reporting budgets using realized $\rho$ (and using $\bar{\rho}$ in tables) when comparing metrics and settings.

### C.3. Entropy-Triggered Practical Variant

The disagreement gate in Section 5 queries both $p_b$ and $p_r$ at every step, which is fine for diagnosis but wasteful at deployment. Since disagreement spikes overlap heavily with base-entropy spikes (Section 4.3, Figure 2c), a cheaper alternative is to trigger on the base entropy $H_b(t) = H(p_b(\cdot | x_t))$ alone, applying the same calibration and gating recipe with $s_t$ replaced by $H_b(t)$. As shown in Table 9, this cheaper trigger already recovers most of the strong-model Pass@8 advantage: at $\bar{\rho} \approx 0.08$, the entropy-guided variant reaches $65.8/87.2$ (Acc/Pass@8), essentially matching Qwen3-8B's Pass@8 of $87.3$ while querying the stronger model at only ~8% of steps.

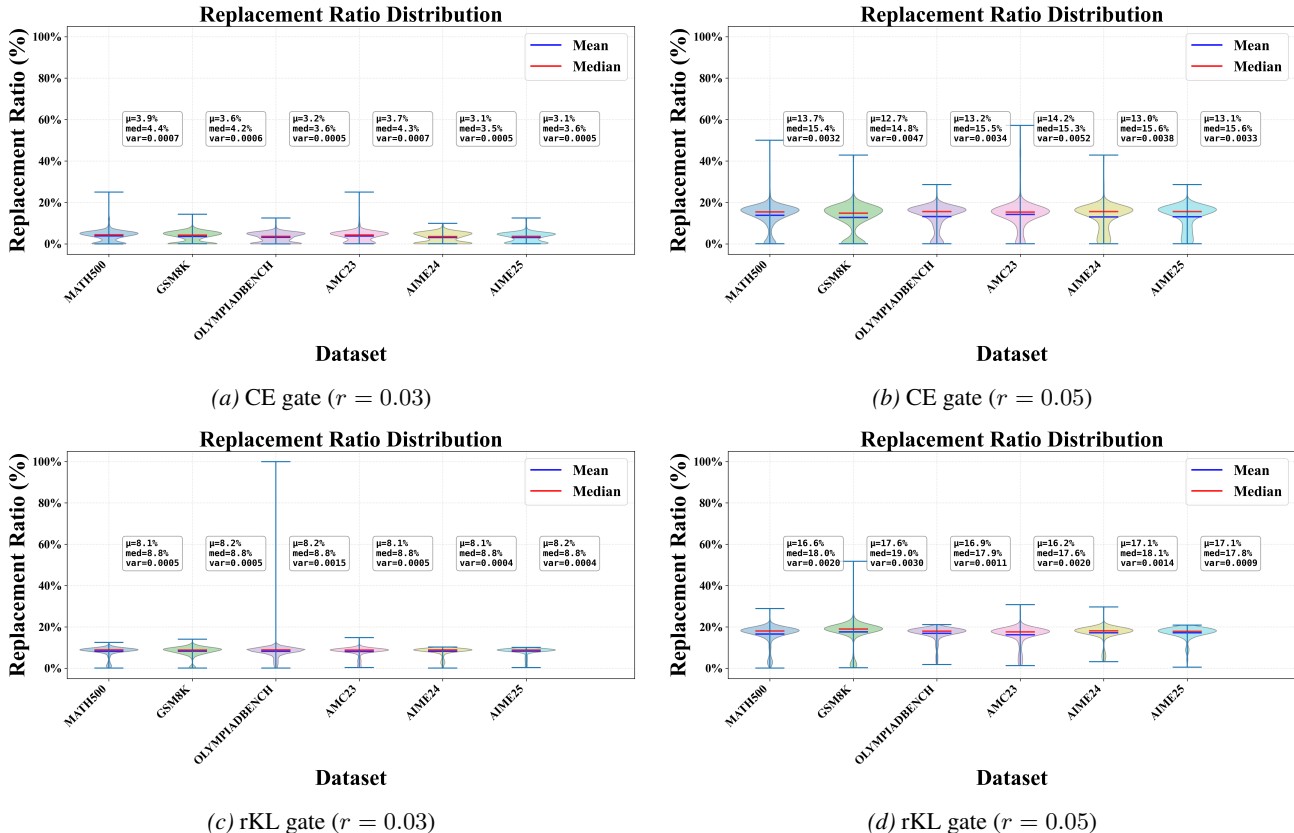

*(a)* CE gate ($r = 0.03$)

*(b)* CE gate ($r = 0.05$)

*(c)* rKL gate ($r = 0.03$)

*(d)* rKL gate ($r = 0.05$)

*Figure 7.* **Stability of realized replacement budget.** Violin plots of per-problem replacement rate $\rho_i$ across benchmarks for Qwen3-0.6B under (top) Cross-Entropy gating and (bottom) reverse-KL gating, each shown at two calibrated spike ratios ($r = 0.03$ and $r = 0.05$). Concentrated distributions indicate that the realized intervention budget is stable across problems rather than dominated by a small subset of outliers.

### C.4. Matched-Budget Baselines: Random and Early-Only

We test whether the gains from disagreement-guided intervention are driven specifically by the choice of which tokens are replaced, by comparing against two budget-matched baselines: (i) a **random** baseline that uniformly samples replacement positions to hit a target budget $\rho$, and (ii) an **early-only** baseline that, given the reasoning model's output of length $L_r$, replaces the first $\rho \cdot L_r$ tokens of the rollout with reasoning-model tokens and then lets the base model continue. The early-only baseline is motivated by our Finding 2 (Section 4.2) that disagreement spikes concentrate at early positions: it isolates whether position alone is sufficient.

Table 10 summarizes the comparison on Qwen3-0.6B (averaged over the six math benchmarks). Two findings stand out:

**Random replacement is consistently dominated by guided.** At an aggressive $25\%$ random replacement, average performance only reaches 26.4/55.2 (Acc/Pass@8), whereas the disagreement-guided gate at only $\bar{\rho} \approx 0.04$ already reaches 29.1/61.4. This confirms that gains do not come merely from inserting stronger-model tokens at arbitrary positions.

**Position alone is not sufficient.** The early-only baseline does improve over the base, consistent with our finding that critical tokens concentrate early. However, even at $\bar{\rho} \approx 0.25$ early-only reaches only 25.7/58.4, which is still worse than guided at $\bar{\rho} \approx 0.04$ (29.1/61.4). This shows that within the early region, only the disagreement / uncertainty spikes are truly trajectory-setting; blindly replacing all early tokens wastes budget on non-critical positions.

Together, these baselines support the interpretation that the gain comes from selecting positions that are simultaneously early and informative, i.e., positions where the base and reasoning models disagree most under base-model uncertainty.

*Table 9.* **Entropy-triggered practical variant (Qwen3-0.6B).** Replacing the cross-model disagreement trigger with base-model entropy $H_b(t)$ already recovers most of the strong-model Pass@8 advantage while only requiring $p_b$ at every step.

| Method | Avg ACC | Avg Pass@8 |
|---|---|---|
| Qwen3-0.6B-Base | 12.9 | 36.0 |
| + Entropy-Guided ($\bar{\rho} \approx 0.08$) | 65.8 | 87.2 |
| + Entropy-Guided ($\bar{\rho} \approx 0.20$) | 71.3 | 86.4 |
| Qwen3-0.6B (Thinking) | 43.4 | 64.1 |
| Qwen3-8B (Thinking) | 78.1 | 87.3 |

*Table 10.* **Matched-budget baselines on Qwen3-0.6B.** Average Accuracy / Pass@8 over the six math reasoning benchmarks. Random samples replacement positions uniformly under a target budget $\rho$. Early-only replaces the first $\rho \cdot L_r$ tokens (where $L_r$ is the reasoning-model rollout length) and then lets the base model continue. Disagreement-guided intervention at $\bar{\rho} \approx 0.04$ already outperforms both baselines at 25% budget.

| Method | Budget $\rho$ | Avg ACC | Avg Pass@8 |
|---|---|---|---|
| Qwen3-0.6B-Base | 0.00 | 13.0 | 36.0 |
| + Random | 0.10 | 17.7 | 42.0 |
| + Random | 0.15 | 20.1 | 46.0 |
| + Random | 0.20 | 21.4 | 50.0 |
| + Random | 0.25 | 26.4 | 55.2 |
| + Early-only | 0.10 | 20.9 | 44.1 |
| + Early-only | 0.15 | 20.9 | 49.9 |
| + Early-only | 0.20 | 23.3 | 53.4 |
| + Early-only | 0.25 | 25.7 | 58.4 |
| + Guided (Ours) | $\bar{\rho} \approx 0.04$ | 29.1 | 61.4 |
| + Guided (Ours) | $\bar{\rho} \approx 0.13$ | 52.4 | 80.0 |
| Qwen3-0.6B (Thinking) | – | 43.4 | 64.1 |
| Qwen3-8B (Thinking) | – | 78.1 | 87.3 |

## C.5. Cross-Family Generalization on LLaMA

To test whether the sparse-control finding transfers beyond the Qwen3 family, we evaluate the disagreement-guided intervention on a LLaMA-based base/guide pair: the base model is LLaMA-3.1-8B (Llama Team, 2024) and the reasoning guide is DeepSeek-R1-Distill-Llama-8B (Guo et al., 2025). Calibration follows the same procedure as in the main experiments (held-out GSM8K-style prompts, sliding window $W = 64$).

Table 11 reports the average performance over the six math benchmarks. The same monotonic recovery pattern holds: with $\bar{\rho} \approx 0.20$, guided intervention recovers ~91% of the gap (in Pass@8) between the base model and the reasoning guide, mirroring the Qwen3 results in Table 3. This indicates that the disagreement-spike phenomenon and the resulting recovery curve are not specific to a single model family or training pipeline.

*Table 11.* **Cross-family generalization on the LLaMA pair (Accuracy / Pass@8).** Base model: LLaMA-3.1-8B; reasoning guide: DeepSeek-R1-Distill-Llama-8B. The Recovery column quantifies how much the intervention recovers reasoning capabilities, calculated as $(P_{Guided} - P_{Base})/(P_{Guide} - P_{Base})$, where $P$ denotes Pass@8 (consistent with Table 3).

| Model | Math500 | GSM8K | AMC23 | Olympiad | AIME24 | AIME25 | Average | Recovery |
|---|---|---|---|---|---|---|---|---|
| LLaMA-3.1-8B (Base) | 6.0 / 30.0 | 16.9 / 66.0 | 2.8 / 17.5 | 1.1 / 7.0 | 0.8 / 6.7 | 0.0 / 0.0 | 4.6 / 21.2 | - |
| + Guided ($\bar{\rho} \approx 0.03$) | 19.4 / 64.0 | 41.8 / 87.0 | 6.2 / 30.0 | 3.2 / 12.0 | 1.2 / 6.7 | 0.4 / 3.3 | 12.0 / 33.8 | 21% |
| + Guided ($\bar{\rho} \approx 0.12$) | 55.5 / 90.0 | 70.9 / 97.0 | 49.4 / 90.0 | 26.2 / 54.0 | 14.2 / 43.3 | 11.7 / 33.3 | 38.0 / 67.9 | 78% |
| + Guided ($\bar{\rho} \approx 0.20$) | 74.6 / 97.0 | 86.2 / 97.0 | 72.5 / 92.5 | 41.2 / 60.0 | 35.4 / 66.7 | 22.5 / 43.3 | 55.4 / 76.1 | 91% |
| DeepSeek-R1-Distill-Llama-8B (Guide) | 86.9 / 98.0 | 90.6 / 97.0 | 86.2 / 95.0 | 51.9 / 65.0 | 42.9 / 80.0 | 30.0 / 53.3 | 64.8 / 81.4 | - |

## C.6. Cross-Domain Generalization to Non-math Scientific QA

To assess whether the sparse-intervention effect transfers beyond mathematical reasoning, we evaluate on GPQA-Diamond (Rein et al., 2024), a benchmark of graduate-level scientific question answering across biology, physics, and

chemistry. We use a 50-question subset and keep the same Qwen3-0.6B-Base / Qwen3-8B (Thinking) pair as in our main experiments.

Table 12 shows two patterns that differ from the math setting. First, same-size reasoning post-training provides little advantage here: Qwen3-0.6B (Thinking) matches the base in Accuracy (23.0 vs. 22.5) and underperforms it in Pass@8 (68.0 vs. 78.0); even the 8B reasoning model does not improve Pass@8 over the 0.6B base (both 78.0). Second, sparse intervention nonetheless yields clear gains: at $\bar{\rho} \approx 0.05$, guided decoding reaches 37.1 / 88.0, improving the base by +14.6 Accuracy and +10.0 Pass@8, and exceeding the 8B reasoning model's Pass@8 (88.0 vs. 78.0). The remaining Accuracy gap to 8B (37.1 vs. 63.2) is larger than on math benchmarks, suggesting that fully closing the non-math reasoning gap may require denser intervention. Together with the cross-family LLaMA results in Appendix C.5, this indicates the sparse-control phenomenon is not specific to a single model family or to mathematical reasoning.

*Table 12.* **Cross-domain generalization on GPQA-Diamond (50-question subset; Accuracy / Pass@8).** Base: Qwen3-0.6B-Base; reasoning guide: Qwen3-8B (Thinking). Same-size reasoning post-training provides little Pass@8 advantage in this domain, while sparse guided intervention improves both Accuracy and Pass@8 over the base and exceeds the 8B reasoning model's Pass@8.

| Model | Accuracy | Pass@8 |
|---|---|---|
| Qwen3-0.6B-Base | 22.5 | 78.0 |
| Qwen3-0.6B (Thinking) | 23.0 | 68.0 |
| + Guided ($\bar{\rho} \approx 0.01$) | 18.8 | 82.0 |
| + Guided ($\bar{\rho} \approx 0.03$) | 26.1 | 78.0 |
| + Guided ($\bar{\rho} \approx 0.05$) | **37.1** | **88.0** |
| Qwen3-8B (Thinking) | 63.2 | 78.0 |

## C.7. Calibration Threshold Stability across Datasets

A natural concern with calibrated gating is whether the calibrated thresholds $\tau$ and the tail-to-mean factor $\lambda$ (Equation (4)) are dataset-specific or transferable. To assess this, we re-calibrate the gate on each of the six benchmarks separately and compare the resulting $(\tau, \lambda)$ at three target spike ratios $r \in \{0.01, 0.03, 0.05\}$. Table 13 shows that: (i) for a fixed $r$, the calibrated $\tau$ values stay within a narrow range across calibration sets (e.g., for $r = 0.01$, $\tau \in [3.18, 4.12]$), and (ii) both $\tau$ and $\lambda$ follow the same monotonic decreasing trend in $r$ across all datasets. This indicates that the disagreement tail used by our gate is reasonably stable, rather than strongly dataset-specific, which is consistent with the calibration recipe used throughout the main paper (a single calibration set is used for all evaluations).

*Table 13.* **Calibration threshold stability across datasets.** For each calibration set we report $\tau$ / $\lambda$ at three target spike ratios $r \in \{0.01, 0.03, 0.05\}$. Values stay within a narrow range across datasets at fixed $r$, and follow the same monotonic trend with $r$, suggesting that the calibrated tail of the disagreement distribution is stable rather than dataset-specific.

| Calibration set | $\tau$ / $\lambda$ ($r = 0.01$) | $\tau$ / $\lambda$ ($r = 0.03$) | $\tau$ / $\lambda$ ($r = 0.05$) |
|---|---|---|---|
| MATH500 | 3.649 / 23.378 | 1.095 / 7.018 | 0.372 / 2.381 |
| GSM8K | 3.667 / 24.443 | 0.467 / 3.114 | 0.239 / 1.591 |
| OlympiadBench | 3.509 / 24.046 | 0.997 / 6.834 | 0.293 / 2.011 |
| AMC23 | 3.630 / 19.675 | 1.164 / 6.308 | 0.539 / 2.919 |
| AIME24 | 3.181 / 27.816 | 0.772 / 6.751 | 0.187 / 1.635 |
| AIME25 | 4.121 / 23.361 | 1.306 / 7.406 | 0.417 / 2.363 |

## C.8. Intervention Harm Analysis (Pass@8 Flip Statistics)

A natural concern is whether the intervention, while improving aggregate performance, can also degrade specific samples that the base model would have answered correctly without help. To quantify this, we examine sample-level Pass@8 label flips between the base model and the guided model on Qwen3-0.6B (with Qwen3-8B as the guide), at $\bar{\rho} \approx 0.13$. For each problem we compare whether any of the 8 sampled trajectories under the base setting and under the guided setting are correct, and we tally three outcomes: **Error→Correct** (the guided setting now has at least one correct sample, while the base did not), **Correct→Error** (the base had at least one correct sample, but the guided does not), and unchanged (both correct or both incorrect; not shown in the table).

Table 14 shows that the dominant effect of intervention is error correction: out of 400 examples, 152 examples transition

*Table 14.* **Sample-level Pass@8 flips on Qwen3-0.6B (guided vs. base, $\bar{\rho} \approx 0.13$).** Across 400 examples, the intervention overwhelmingly converts errors to successes (152 flips), while only 3 cases flip in the opposite direction.

| Dataset | #Examples | Error→Correct | Correct→Error |
|---|---|---|---|
| AIME24 | 30 | 20 | 0 |
| AIME25 | 30 | 17 | 0 |
| AMC23 | 40 | 20 | 1 |
| GSM8K | 100 | 18 | 1 |
| MATH500 | 100 | 32 | 0 |
| OlympiadBench | 100 | 45 | 1 |
| **Total** | **400** | **152** | **3** |

from incorrect to correct under guided decoding, while only 3 transition in the opposite direction (a ∼50:1 ratio). The three Correct→Error cases are spread across AMC23, GSM8K, and OlympiadBench, and we observe no systematic pattern: each is a case where the base model had a borderline-correct trajectory that the intervention diverted toward a different (incorrect) chain. This tail behavior is consistent with the fact that intervention edits the trajectory at a small set of high-uncertainty decision points: in rare cases, redirecting these decisions can move a borderline-correct rollout off-track.

## D. Case Study

We visualize the divergence-guided takeover on this example. Figure 8 shows the prompt, and Figure 9 provides the full guided rollout.

**What gets replaced.** A salient pattern is that most replaced tokens occur in planning- and control-related regions of the response: the brief outline of the solution, the choice of intermediate variables, and the decision of which quantities to compute next. In contrast, once the correct plan is established, the subsequent arithmetic manipulations and execution steps are largely produced by the base model itself, with few (or no) additional takeovers.

**Interpretation.** This example supports the view that the base–reasoning gap is not primarily due to missing low-level execution capability. Rather, failures concentrate at a small number of trajectory-defining decisions—what to do next and how to structure the computation—where the base model tends to commit to an unproductive path. By selectively overriding these planning tokens, the takeover effectively supplies a lightweight "controller" that keeps the rollout on-track, while leaving the base model to carry out the bulk of the calculation.

**Connection to our main findings.** Consistent with our sparsity analysis, the interventions are localized and concentrated, yet have outsized downstream effects: correcting a few high-disagreement planning choices is sufficient to steer the remainder of the trajectory toward a correct solution.

> **Problem:** In the first half of a soccer match, team A scores 4 goals while team B scores 2 goals fewer than team A. In the second half, team A scores 1/4 of the number of goals scored by team B, which scores 4 times the number of goals it scored in the first half. What's the total number of goals scored in the match?

*Figure 8.* **Case study prompt.** The following figure visualize the token-level takeover trajectory for this example.

<think>
Okay, let's try to figure out the total number of goals scored in the match. Hmm, let's break it down step by step. First, the problem mentions two halves of the match: the first half and the second half. Let's start with the first half.

In the first half, team A scored 4 goals. Then, the problem says that team B scored 2 goals fewer than team A. So, team B scored 4 − 2 = 2 goals in the first half. Wait, that seems straightforward.

Now, moving to the second half. The problem states that team A scores 1/4 of the number of goals scored by team B in the second half. But here, it says that team B scores 4 times the number of goals it scored in the first half.

Wait, that seems important. Let me double-check that. Team B's goals in the first half are 2. Then, in the second half, team B scores 4 times that. So, team B scores 2 * 4 = 8 goals in the second half.

Now, we need to find out how many goals team A scored in the second half. The problem says that team A scores 1/4 of the number of goals scored by team B in the second half. So, team A's second half goals would be 1/4 of team B's second half goals.

Team B's second half goals are 8, so team A's second half goals would be 8 * (1/4) = 2 goals.

Okay, so let's summarize:
First half:
– Team A: 4 goals
– Team B: 2 goals
Second half:
– Team B: 8 goals
– Team A: 2 goals

Now, to find the total number of goals scored in the match, we need to add up all the goals from both halves for both teams.

Total goals by Team A: 4 (first half) + 2 (second half) = 6 goals.
Total goals by Team B: 2 (first half) + 8 (second half) = 10 goals.

Therefore, the total number of goals scored in the match is 6 + 10 = 16 goals.

Wait, let me check again to make sure I didn't make a mistake.

First half: Team A 4, Team B 2. That's correct because Team B scored 2 fewer than Team A.

Second half: Team B scores 4 times the first half, so 4 * 2 = 8.

Team A scores 1/4 of Team B's second half goals, so 1/4 * 8 = 2.

Total for Team A: 4 + 2 = 6 goals.
Total for Team B: 2 + 8 = 10 goals.
Total goals: 6 + 10 = 16.

Hmm, that seems correct. Let me check if there's any other way to interpret the problem.

Wait, the problem says "team A scores 1/4 of the number of goals scored by team B, which scores 4 times the number of goals it scored in the first half."

Let me parse that sentence again.

Team B's goals in the second half are 4 times the number of goals it scored in the first half. So, Team B's second half goals are 4 * 2 = 8.

Team A scores 1/4 of Team B's second half goals, so 1/4 * 8 = 2.

Yes, that's the same as before.

So, adding up all the goals, Team A has 4 + 2 = 6, Team B has 2 + 8 = 10. Total is 6 + 10 = 16.

Therefore, the total number of goals scored in the match is 16.

Wait, I think that's correct. Let me check once more.

First half: Team A 4, Team B 2.
Second half: Team B 8, Team A 2.
Total: 4 + 2 + 8 + 2 = 16.

Yes, that's right.

So, the answer should be 16.

**Final Answer**
The total number of goals scored in the match is \(\boxed{16}\).

But wait, the problem says "This is the problem: In the first half of a soccer match, team A scores 4 goals while team B scores 2 goals fewer than team A. In the second half, team A scores 1/4 of the number of goals scored by team B, which scores 4 times the number of goals it scored in the first half. What's the total number of goals scored in the match?"

Wait, the wording for the second half is a bit confusing. Let me check again.

"In the second half, team A scores 1/4 of the number of goals scored by team B, which scores 4 times the number of goals it scored in the first half."

So, the phrase "which scores 4 times the number of goals it scored in the first half" refers to team B.

So, Team B scores 4 times the number of goals it scored in the first half during the second half.

So, Team B's second half goals are 4 * 2 = 8.

Then, Team A scores 1/4 of Team B's second half goals, so 1/4 * 8 = 2.

So, that's the same as before.

Therefore, the total is 4 + 2 + 8 + 2 = 16.

Yes, that's correct.

So, the answer is 16.

**Final Answer**
\boxed{16}

But wait, the problem says "This is the problem: In the first half of a soccer match, team A scores 4 goals while team B scores 2 goals fewer than team A. In the second half, team A scores 1/4 of the number of goals scored by team B, which scores 4 times the number of goals it scored in the first half. What's the total number of goals scored in the match?"

Wait, maybe there's a different interpretation. Let me check again.

"In the second half, team A scores 1/4 of the number of goals scored by team B, which scores 4 times the number of goals it scored in the first half."

The wording here is a bit ambiguous. Does "which" refer to team B?

Yes, because "team B, which scores 4 times the number of goals it scored in the first half."

So, Team B's second half goals are 4 times its first half goals, which is 4 * 2 = 8.

Then, Team A scores 1/4 of Team B's second half goals, so 8 * 1/4 = 2.

So, that's the same as before.

So, the total is 4 + 2 + 8 + 2 = 16.

Yes, so the answer is 16.

**Final Answer**
\boxed{16}
</think>

To determine the total number of goals scored in the match, we need to analyze the information provided step by step.

---

### Step 1: First Half Goals
– Team A scores **4** goals.
– Team B scores **2** goals fewer than Team A.

Calculation:
$$
\text{Team B's first half goals} = \text{Team A's first half goals} - 2 = 4 - 2 = 2
$$
---

### Step 2: Second Half Goals
– Team B scores **4 times** the number of goals it scored in the first half.

Calculation:
$$
\text{Team B's second half goals} = 4 \times 2 = 8
$$
– Team A scores **1/4** of the number of goals Team B scored in the second half.

Calculation:
$$
\text{Team A's second half goals} = \frac{1}{4} \times 8 = 2
$$
---

### Step 3: Total Goals
Now, sum up all the goals scored by both teams in both halves.
– **Team A total:**
$$
\text{First half} + \text{Second half} = 4 + 2 = 6
$$
– **Team B total:**
$$
\text{First half} + \text{Second half} = 2 + 8 = 10
$$
– **Total goals scored in the match:**
$$
\text{Team A total} + \text{Team B total} = 6 + 10 = 16
$$
---

### Final Answer
$$
\boxed{16}
$$

*Figure 9.* **Case study rollout.** Highlighted tokens denote reasoning model replacements triggered by disagreement spikes.