# OpenReview forum: "Reasoning Can Be Restored by Correcting a Few Decision Tokens"
_ICML.cc/2026/Conference — ICML 2026 regular_

### Official Review · Reviewer_EYDx · 2026-03-05

**Soundness:** 2
**Presentation:** 2
**Significance:** 2
**Originality:** 2
**Overall Recommendation:** 4
**Confidence:** 4

**Summary:**

This paper studies the token-level disagreement between base LLMs and (LRMs). The authors find that the disagreement is highly sparse and mostly occurs at early “decision tokens” related to planning. Based on this observation, they propose a disagreement-guided intervention strategy that temporarily delegates token generation to a stronger reasoning model when disagreement exceeds a threshold.

**Compliance With Llm Reviewing Policy:**

Affirmed.

**Final Justification:**

Please see my response to AC :)

**Key Questions For Authors:**

1. Does the sparse disagreement pattern hold for other model families (e.g., LLaMA)?
2. Are the identified planning tokens causally important for reasoning, or just correlated with successful solutions?
3. What are the actual latency and compute costs compared to running a reasoning model directly?
4. Are there cases where the intervention hurts performance?

**Limitations:**

Yes.

**Strengths And Weaknesses:**

# Strengths

- Interesting observation. The paper shows that base–LRM disagreement is sparse and concentrated early in generation.
- Simple method. The disagreement-based token intervention is easy to implement and requires no training.
- Empirical gains. Replacing a small fraction of tokens can significantly improve reasoning accuracy on several math benchmarks.

# Weaknesses

- Limited scope. Experiments are restricted to Qwen3 models and math tasks. It is unclear whether the findings generalize to other architectures or domains.
- Heuristic planning tokens. Planning tokens are identified using keyword rules, which makes the “planning enrichment” result somewhat assumption-dependent.
- Unclear mechanism. The paper shows performance recovery but does not clearly explain why the intervention works.
- Efficiency claims are incomplete. The method relies on a larger reasoning model, but latency and compute costs are not fully analyzed.
- Calibration required. The disagreement threshold must be tuned, and the intervention rate varies across datasets.

---

> ### Author Rebuttal · Authors · 2026-03-31
>
> We sincerely appreciate your careful review and helpful suggestions. We respond to your comments as follows.
>
> > **W1 & Q1: Generalization across models and tasks**
>
> For cross-family generalization, we tested a LLaMA-based pair (LLaMA-3.1-8B as base; DeepSeek-R1-Distill-Llama-8B as guide) [2]:
>
> |Model|Average|Recovery|
> |---|---|---:|
> |LLaMA-3.1-8B|4.6/21.2|-|
> |+Guided($\bar{\rho}\approx0.03$)|12.0/33.3|20%|
> |+Guided($\bar{\rho}\approx0.12$)|38.0/67.9|78%|
> |+Guided($\bar{\rho}\approx0.20$)|55.4/76.1|91%|
> |DeepSeek-R1-Distill-Llama-8B|64.8/81.4|-|
>
> These results suggest that the sparse-intervention effect is not limited to Qwen. To broaden the scope beyond math, we also report performance on GPQA-Diamond [1]:
>
> |Model|GPQA-Diamond(ACC/Pass@8)|
> |---|---:|
> |Qwen3-0.6B-Base|0.0/0.0|
> |+Guided($\bar{\rho}\approx0.01$)|18.8/82.0|
> |+Guided($\bar{\rho}\approx0.06$)|26.1/78.0|
> |+Guided($\bar{\rho}\approx0.13$)|37.1/88.0|
> |Qwen3-0.6B(Thinking)|3.5/22.0|
> |Qwen3-8B|41.5/68.0|
>
> Guided intervention substantially improves performance on GPQA-Diamond, suggesting that the sparse-control picture also transfers beyond math. We will expand to additional tasks beyond math in the next version.
>
> > **W2 & Q2: Informative role of planning-related tokens**
>
> Our planning-token analysis is intended as a simple and transparent probe into where disagreement concentrates, while the intervention results directly test whether selected positions are informative. To test whether the intervention positions are meaningful rather than arbitrary, we added a random baseline:
>
> |Method|Rate|Avg ACC|Avg Pass@8|
> |---|---:|---:|---:|
> |Base|0.00|13.0|36.0|
> |+Random|0.10|17.7|42.0|
> |+Random|0.15|20.1|46.0|
> |+Random|0.20|21.4|50.0|
> |+Random|0.25|26.4|55.2|
> |+Guided|$\bar{\rho}\approx0.04$|29.1|61.4|
> |+Guided|$\bar{\rho}\approx0.13$|52.4|80.0|
>
> Guided intervention at only $\bar{\rho}\approx 0.04$ already outperforms even $25\%$ random replacement. This supports the view that the gain comes from selecting informative positions, not arbitrary insertion.
>
> > **W3: Mechanistic explanation of sparse intervention**
>
> Our interpretation: (1) disagreement is sparse, so only a few positions create most of the base–reasoning gap; (2) the largest spikes occur early, when the model is still choosing how to structure the solution; and (3) correcting early decisions redirects the downstream rollout, as confirmed by the random baseline.
>
> > **W4 & Q3: Practical efficiency and compute overhead**
>
> Thank you for raising this practical question. Our main contribution is to show that the base–reasoning gap is highly sparse and concentrated in few critical decision tokens, and the intervention validates that finding. For practical usage, the full disagreement trigger can be replaced by a cheaper uncertainty-based proxy.
>
> |Method|Avg ACC|Avg Pass@8|
> |---|---:|---:|
> |Qwen3-0.6B-Base|12.9|36.0|
> |+Entropy-Guided($\bar{\rho}\approx0.08$)|65.8|87.2|
> |+Entropy-Guided($\bar{\rho}\approx0.20$)|71.3|86.4|
> |Qwen3-8B|78.1|87.3|
>
> These results show that the sparse-control idea is not tied to a full cross-model disagreement trigger: a cheaper uncertainty-based trigger already recovers most of the strong model’s Pass@8.
>
> > **W5: Calibration robustness**
>
> The gate does require calibration, but we find that the calibrated thresholds are stable across datasets. To check this, we compared the resulting $\tau$ and $\lambda$ on several datasets under the same target $\alpha=0.01, 0.03, 0.05$, the operating regimes used in our experiments.
>
> |Calibration set|$\alpha=0.01$($\tau/\lambda$)|$\alpha=0.03$($\tau/\lambda$)|$\alpha=0.05$($\tau/\lambda$)|
> |---|---:|---:|---:|
> |MATH500|3.649/23.378|1.095/7.018|0.372/2.381|
> |GSM8K|3.667/24.443|0.467/3.114|0.239/1.591|
> |OlympiadBench|3.509/24.046|0.997/6.834|0.293/2.011|
> |AMC23|3.630/19.675|1.164/6.308|0.539/2.919|
> |AIME24|3.181/27.816|0.772/6.751|0.187/1.635|
> |AIME25|4.121/23.361|1.306/7.406|0.417/2.363|
>
> The calibration is reasonably stable rather than arbitrarily tuned to a single dataset: While the exact values are not identical, they remain in the same range and follow the same monotonic trend as $\alpha$ increases. A fuller downstream analysis will appear in the appendix.
>
> > **Q4: When intervention hurts**
>
> We checked Pass@8 sample-level label flips (0.6B as base, 8B as guide). The dominant pattern is error-to-correct, while correct-to-error cases are rare:
>
> |Dataset|#Examples Compared|Error→Correct|Correct→Error|
> |---|---:|---:|---:|
> |AIME24|30|20|0|
> |AIME25|30|17|0|
> |AMC23|40|20|1|
> |GSM8K|100|18|1|
> |MATH500|100|32|0|
> |OlympiadBench|100|45|1|
> |Total|400|152|3|
>
> Intervention overwhelmingly converts errors to successes (152 vs. 3 flips). In practice, the dominant effect is error correction rather than performance degradation.
>
> [1] Rein et al. GPQA: A Graduate-Level Google-Proof Q&A Benchmark. ICLR 2024.
>
> [2] Grattafiori et al. The Llama 3 Herd of Models. arXiv 2024.

---

> > ### Author Rebuttal · Reviewer_EYDx · 2026-04-02
> >
> > Thank you for providing the additional experiments and clarifications, and my concerns have been addressed. After careful consideration, I will maintain my current score.

---

> > > ### Author Response · Authors · 2026-04-03
> > >
> > > Thank you again for your timely reply. We are very grateful that you found the main concerns fully addressed. May I kindly know the reason for keeping the scores? As we have addressed all your concerns.
> > >
> > > We completely respect your careful consideration. Since the additional experiments and clarifications appear to have resolved the issues you raised, we would be very grateful if you could reconsider whether the updated evidence changes your overall assessment of the paper. In particular, we hope the new results on cross-family generalization, non-math evaluation, informative-position baselines, and robustness help strengthen the case that the paper's core finding is both substantive and broadly relevant.
> > >
> > > If there are aspects that you feel still limit the paper's contribution despite the resolved concerns, we would really appreciate hearing them so we can further improve. We sincerely appreciate your time and detailed feedback.

---

### Official Review · Reviewer_y2tW · 2026-03-13

**Soundness:** 2
**Presentation:** 4
**Significance:** 3
**Originality:** 2
**Overall Recommendation:** 5
**Confidence:** 4

**Summary:**

The paper investigates the performance gap and the qualitative token-level differences between base language models and large reasoning models. The authors measure the cross-entropy between the next-token distributions of a base and a reasoning model. They find that: (1) disagreement is concentrated in few tokens, (2) it occurs primarily early in the rollout, (3) disagreement spikes coincide with base-model uncertainty spikes and enriched for planning-related tokens, (4) it is predictive of reasoning failures. Based on these observations, they propose an intervention scheme, where they generate a single token using the reasoning model only at high-disagreement positions, while using the base model for everything else. On math benchmarks with the Qwen3 family of models, intervening on 8-13% of tokens recovers (or sometimes even exceeds) the performance of a reasoning model of the same size.

**Compliance With Llm Reviewing Policy:**

Affirmed.

**Final Justification:**

The paper makes an interesting contribution and I believe that the additional experiments and ablations added by the authors make it a complete and informative study of the phenomenon discussed.

**Key Questions For Authors:**

* The planning vs execution token classifier is very coarse and relies on a simple keyword heuristic. Do you think it would make sense to conduct a sensitivity analysis using some LLM-based classifier that is fed with a sequence of tokens and not just the next token?

* Have you tried running the simple baseline where you randomly select intervention points under the same budget? It would further strengthen the results if you added this as another baseline to compare against.

* In section 2 you mention "dynamically take over generation at divergence peaks to correct reasoning paths (Peng et al., 2025)". This sounds very similar to your intervention scheme. How do they compare?

**Limitations:**

yes

**Strengths And Weaknesses:**

# Strengths

* The paper is very well-written and motivated. It presents clear findings and suggests an interesting intervention that is tested experimentally.

* The finding that a 0.6B model can recover the reasoning capabilities of an 8B LRM by just a sparse single-token interventions is very interesting and it might have important real-world implications, if it scales.

# Weaknesses

* Currently, the experiments testing the intervention scheme require both models to be evaluated at every single token. This is of course not practically relevant. The paper has to include the far more interesting experiment of what happens if we use the base model inherent uncertainty (entropy) as a trigger for generating the next token by the LRM. This should be compared with the proposed method and the other baselines.

* How sensitive are the results to the calibration method? How do they change if you calibrate using different datasets? I think it would be valuable to do this ablation.

* I think the results would be greatly strengthened if you did include additional model families and/or additional reasoning benchmarks. Also, how does performance change if you use different-sized LRMs for the single-token interventions?

## Minor issues

* Section 4 first paragraph references "section 3.2" four times instead of references sections 4.1-4.4

---

> ### Author Rebuttal · Authors · 2026-03-31
>
> We sincerely appreciate your thoughtful review and highly valuable suggestions. We respond to your comments as follows.
>
> > **W1: Entropy-triggered intervention**
>
> We added an entropy-triggered variant that uses only base-model uncertainty to invoke the stronger model, serving as a much cheaper trigger.
>
> |Method|Avg ACC|Avg Pass@8|
> |---|---:|---:|
> |Qwen3-0.6B-Base|12.9|36.0|
> |+Entropy-Guided($\bar{\rho}\approx0.08$)|65.8|87.2|
> |+Entropy-Guided($\bar{\rho}\approx0.20$)|71.3|86.4|
> |Qwen3-0.6B(Thinking)|43.4|64.1|
> |Qwen3-8B|78.1|87.3|
>
> This shows that the sparse-intervention effect is not tied to a full cross-model disagreement trigger.
>
> > **Q2: Random intervention baseline**
>
> We added a matched-budget random baseline.
>
> |Method|Budget|Avg ACC|Avg Pass@8|
> |---|---:|---:|---:|
> |Qwen3-0.6B-Base|0.00|13.0|36.0|
> |+Random|0.10|17.7|42.0|
> |+Random|0.15|20.1|46.0|
> |+Random|0.20|21.4|50.0|
> |+Random|0.25|26.4|55.2|
> |+Guided|$\bar{\rho}\approx0.04$|29.1|61.4|
> |+Guided|$\bar{\rho}\approx0.13$|52.4|80.0|
>
> Notably, guided intervention at only $\bar{\rho}\approx 0.04$ already outperforms 25% random replacement. This supports the claim that gains come from selecting informative positions, not just inserting stronger-model tokens somewhere.
>
> > **W2: Calibration sensitivity**
>
> We compared the calibrated $\tau$ and $\lambda$ obtained from different datasets under the same target $\alpha=0.01, 0.03, 0.05$, which are exactly the operating regimes used in our experiments.
>
> |Calibration set|$\alpha=0.01$($\tau/\lambda$)|$\alpha=0.03$($\tau/\lambda$)|$\alpha=0.05$($\tau/\lambda$)|
> |---|---:|---:|---:|
> |MATH500|3.649/23.378|1.095/7.018|0.372/2.381|
> |GSM8K|3.667/24.443|0.467/3.114|0.239/1.591|
> |OlympiadBench|3.509/24.046|0.997/6.834|0.293/2.011|
> |AMC23|3.630/19.675|1.164/6.308|0.539/2.919|
> |AIME24|3.181/27.816|0.772/6.751|0.187/1.635|
> |AIME25|4.121/23.361|1.306/7.406|0.417/2.363|
>
> The calibrated thresholds are broadly similar across datasets and follow the same monotonic trend as $\alpha$ increases, suggesting that the disagreement tail used by our gate is reasonably stable rather than strongly dataset-specific. A fuller study will appear in the appendix.
>
> > **W3: Generalization across models and tasks**
>
> Here we report the performance of our method in a new model setting and on a general reasoning task beyond math. For cross-family generalization, we tested a LLaMA-based pair (LLaMA-3.1-8B as base; DeepSeek-R1-Distill-Llama-8B as guide) [2]:
>
> |Model|Average|Recovery|
> |---|---|---:|
> |LLaMA-3.1-8B|4.6/21.2|-|
> |+Guided($\bar{\rho}\approx0.03$)|12.0/33.3|20%|
> |+Guided($\bar{\rho}\approx0.12$)|38.0/67.9|78%|
> |+Guided($\bar{\rho}\approx0.20$)|55.4/76.1|91%|
> |DeepSeek-R1-Distill-Llama-8B|64.8/81.4|-|
>
> These results suggest that the sparse-intervention effect is not limited to Qwen.
> To broaden the scope beyond math, we also report performance on GPQA-Diamond [1]:
>
> |Model|GPQA-Diamond(ACC/Pass@8)|
> |---|---:|
> |Qwen3-0.6B-Base|0.0/0.0|
> |+Guided($\bar{\rho}\approx0.01$)|18.8/82.0|
> |+Guided($\bar{\rho}\approx0.06$)|26.1/78.0|
> |+Guided($\bar{\rho}\approx0.13$)|37.1/88.0|
> |Qwen3-0.6B(Thinking)|3.5/22.0|
> |Qwen3-8B|41.5/68.0|
>
> Guided intervention substantially improves performance on GPQA-Diamond, suggesting that the sparse-control effecttransfers beyond math. And we will further expand the evaluation to additional general reasoning tasks.
>
> > **Minor: Cross-reference correction**
>
> Thank you for catching this. We have fixed the references to Sections 4.1–4.4.
>
> > **Q1: Robustness of the planning-token classifier**
>
> Thank you for this helpful suggestion. We intentionally use the keyword heuristic for transparency and reproducibility, avoiding introducing another learned component whose bias would need separate validation. That said, a context-aware LLM-based classifier is a valuable suggestion， which could capture implicit planning cues that keyword matching misses and provide a richer robustness check. We will include an LLM-based sensitivity analysis in the next version.
>
> > **Q3: Comparison to AdaSwitch**
>
> AdaSwitch [3] is primarily for distillation/training, whereas our method targets inference-time reasoning recovery. Mechanistically, AdaSwitch hands control to the teacher for a span after a divergence point, while our method uses strictly interleaved one-token takeover. More importantly, our main contribution is the sparsity analysis; the intervention validates that finding.
>
> [1] Rein et al. GPQA: A Graduate-Level Google-Proof Q&A Benchmark. ICLR 2024.
>
> [2] Grattafiori et al. The Llama 3 Herd of Models. arXiv 2024.
>
> [3] Peng et al. AdaSwitch: Adaptive Switching Generation for Knowledge Distillation. arXiv 2025.

---

> > ### Author Rebuttal · Reviewer_y2tW · 2026-04-03
> >
> > Thank you for adding all these additional extensive ablations!
> > I think they greatly complement and strengthen the results of your paper and I have increased my score.

---

> > > ### Author Response · Authors · 2026-04-03
> > >
> > > We sincerely appreciate your time and valuable suggestions. We are glad that our response has addressed your concerns, and we would include all your suggestions in our latest paper.

---

### Official Review · Reviewer_kuFc · 2026-03-13

**Soundness:** 2
**Presentation:** 3
**Significance:** 2
**Originality:** 2
**Overall Recommendation:** 4
**Confidence:** 3

**Summary:**

The paper proposes an inference-time mechanism such that a system may use a stronger reasoning model only when base-LRM disagreement is large and otherwise uses the base model. This takes advantage of the sparse reasoning advantage and help motivate a token-level view of reasoning. Experiments show that such token-level intervention can have boosted reasoning performance.

**Compliance With Llm Reviewing Policy:**

Affirmed.

**Final Justification:**

The rebuttal addressed my main concerns and I have revised my score to 4 (weak accept). I think the paper has sufficient soundness, originality, significance, and clarity after my concerns were addressed in the rebuttal. I think such an inference-time mechanism is useful and has important gains over various baselines.

**Key Questions For Authors:**

1. Is there a need to use base-LRM disagreement in the intervention process or would just focusing on the early tokens be sufficient?
2. It would be helpful to see an example of some other type of model or task to see how this approach works in different scenarios.
3. How does token-level intervention compare to existing intervention/collaborative decoding methods?
4. What is the computational overhead of calculating these disagreement scores vs just using the stronger model?

**Limitations:**

Yes

**Strengths And Weaknesses:**

**Soundness**

Strengths:
- Clear methodology
- There are experiments across several different mathematical reasoning benchmarks and solid ablation studies

Weaknesses:
- Has some practicality concerns as the intervention framework still requires full usage of the stronger reasoning model
- Does not have strong generalization to other tasks and models

**Presentation**

Strengths:
- Clear figures and good example in Figure 6 to show the difference between Base and Guided

Weaknesses:
- The paper could include more details on the reproducibility

**Significance**

Strengths:
- Addresses an important question about the base-reasoning gap between models and addresses it at the token-level which can provide a more interesting interpretability aspect

Weaknesses:
- Seems more like an incremental system improvement and has a heavy reliance on stronger reasoning models
- Unsure about the practical usage of the approach or how future works may build upon this

**Originality**

Strengths:
- Novel token-level analysis and interpretation of disagreement between base-reasoning models
- New intervention perspective and interesting takeaways on early high disagreement tokens

Weaknesses:
- The method seems pretty similar to past methods, novelty mostly comes from the analysis

---

> ### Author Rebuttal · Authors · 2026-03-31
>
> We sincerely appreciate your constructive feedback. To address your main concerns, we added several targeted experiments, and we respond to your questions as follows.
>
> > **W1 & Q4: Practicality and computational overhead**
>
> Thank you for raising this practical concern. The main point of our paper is that the base–reasoning gap is concentrated in a small number of critical tokens. The disagreement-guided intervention is used to demonstrate this by showing that sparse correction alone can recover large reasoning gains.
>
> To address practical usage more directly, we added an entropy-triggered variant that uses only the base model’s uncertainty to decide when to invoke the stronger model.
>
> |Method|Avg ACC|Avg Pass@8|
> |---|---:|---:|
> |Qwen3-0.6B-Base|12.9|36.0|
> |+Entropy-Guided($\bar{\rho}\approx0.08$)|65.8|87.2|
> |+Entropy-Guided($\bar{\rho}\approx0.20$)|71.3|86.4|
> |Qwen3-0.6B(Thinking)|43.4|64.1|
> |Qwen3-8B|78.1|87.3|
>
> This shows that the same sparse-control idea remains effective even with a much cheaper uncertainty-based trigger.
>
> > **W2 & Q2: Generalization across models and tasks**
>
> To address this, we added experiments in both directions.
>
> For cross-family generalization, we evaluated a different base/guide pair on the LLaMA backbone [2]:
>
> |Model|Average|Recovery|
> |---|---|---:|
> |LLaMA-3.1-8B|4.6/21.2|-|
> |+Guided($\bar{\rho}\approx0.03$)|12.0/33.3|20%|
> |+Guided($\bar{\rho}\approx0.12$)|38.0/67.9|78%|
> |+Guided($\bar{\rho}\approx0.20$)|55.4/76.1|91%|
> |DeepSeek-R1-Distill-Llama-8B|64.8/81.4|-|
>
> For non-math evaluation, we added GPQA-Diamond [1]:
>
> |Model|GPQA-Diamond(ACC/Pass@8)|
> |---|---:|
> |Qwen3-0.6B-Base|0.0/0.0|
> |+Guided($\bar{\rho}\approx0.01$)|18.8/82.0|
> |+Guided($\bar{\rho}\approx0.06$)|26.1/78.0|
> |+Guided($\bar{\rho}\approx0.13$)|37.1/88.0|
> |Qwen3-0.6B(Thinking)|3.5/22.0|
> |Qwen3-8B|41.5/68.0|
>
> These results suggest that the sparse-intervention effect is not limited to Qwen or to mathematical reasoning.
>
> > **Q1: Necessity of disagreement-based triggering**
>
> Due to the rebuttal-time limit, we prioritized two ablations that more directly test whether the trigger is informative: a matched-budget random baseline and the entropy-triggered variant above.
>
> |Method|Budget/Rate|Avg ACC|Avg Pass@8|
> |---|---:|---:|---:|
> |Qwen3-0.6B-Base|0.00|13.0|36.0|
> |+Random|0.10|17.7|42.0|
> |+Random|0.15|20.1|46.0|
> |+Random|0.20|21.4|50.0|
> |+Random|0.25|26.4|55.2|
> |+Guided|$\bar{\rho}\approx0.04$|29.1|61.4|
> |+Guided|$\bar{\rho}\approx0.13$|52.4|80.0|
>
>
> Guided intervention at only $\bar{\rho}\approx 0.04$ already outperforms 25% random replacement. This shows that the gains do not come merely from inserting stronger-model tokens arbitrarily, but from selecting informative positions. The early-only baseline is also a useful direction, and we will include it in a follow-up version.
>
> > **Q3: Comparison to prior intervention methods**
>
> Thank you for this helpful question. Our method is closest in spirit to token-level takeover methods, but its main contribution is different: the paper’s novelty lies in the analysis showing that the base–reasoning gap is sparse, early, and planning-centered, while the intervention serves as a validation mechanism for that finding. We have clarified this distinction in the related-work discussion. Compared with AdaSwitch-style methods, our setting is inference-time reasoning recovery rather than distillation, and our mechanism is a strictly interleaved one-token takeover rather than handing off an extended span after a divergence point [3].
>
> > **W3: Reproducibility details**
>
> Thank you for pointing this out. In the revised version, we will expand the appendix to include a fuller description of the experimental setup. We will also release the full codebase to reproduce our experiments end-to-end.
>
> [1] Rein et al. GPQA: A Graduate-Level Google-Proof Q&A Benchmark. ICLR 2024.
>
> [2] Grattafiori et al. The Llama 3 Herd of Models. arXiv 2024.
>
> [3] Peng et al. AdaSwitch: Adaptive Switching Generation for Knowledge Distillation. arXiv 2025.

---

> > ### Author Rebuttal · Reviewer_kuFc · 2026-04-01
> >
> > Thanks for the response, it makes sense that the entropy-guided version would be better than the random baseline, but I would like to see if the early-only baseline is comparable to the entropy-guided one as that would be a much simpler way to do the replacement.

---

> > > ### Author Response · Authors · 2026-04-03
> > >
> > > Thank you for this valuable follow-up. We ran the early-only baseline, which first generates a full response using the reasoning model to obtain its length $T$, then replaces the first $\rho \cdot T$ tokens with reasoning-model outputs and lets the base model continue the rest, without any disagreement or uncertainty signal:
> > >
> > > |Model|Average ACC|Average Pass@8|
> > > |---|---:|---:|
> > > |Qwen3-0.6B-Base|12.9|36.0|
> > > |+Early-only($\bar{\rho}\approx0.15$)|20.9|44.1|
> > > |+Early-only($\bar{\rho}\approx0.22$)|20.9|49.9|
> > > |+Early-only($\bar{\rho}\approx0.28$)|23.3|53.4|
> > > |+Early-only($\bar{\rho}\approx0.35$)|25.7|58.4|
> > > |+Guided ($\bar{\rho}\approx0.04$)|29.1|61.4|
> > > |+Guided ($\bar{\rho}\approx0.13$)|52.4|80.0|
> > > |Qwen3-0.6B(Thinking)|43.4|64.1|
> > > |Qwen3-8B|78.1|87.3|
> > >
> > > Early-only does improve over the base, consistent with our Finding 2 that critical tokens concentrate early. However, Guided at only $\bar{\rho}\approx0.04$ already outperforms Early-only at $\bar{\rho}\approx0.35$ in both ACC and Pass@8. This shows that position alone is not sufficient — within the early region, only the disagreement/uncertainty spikes are truly trajectory-setting. Blindly replacing all early tokens wastes budget on non-critical positions. This confirms that base-LRM disagreement，or entropy as a cheaper proxy，is indeed necessary for efficient intervention. Focusing on early tokens alone requires a substantially larger replacement budget to achieve comparable recovery.
> > >
> > > Per-dataset results will be included in the revised paper. We sincerely appreciate your suggestions and will incorporate all feedback into the next version.

---

### Official Review · Reviewer_5ER5 · 2026-03-13

**Soundness:** 3
**Presentation:** 2
**Significance:** 2
**Originality:** 2
**Overall Recommendation:** 4
**Confidence:** 3

**Summary:**

This paper analyzes why reasoning models outperform base language models and finds that the performance gap is highly sparse: most differences arise from a small number of early planning tokens where key reasoning decisions occur. Based on this insight, the authors propose the concept and method of disagreement-guided intervention, where a stronger reasoning model generates a token only when its prediction strongly disagrees with the base model. Experiments across math benchmarks show that intervening on a small fraction of tokens recovers much of the reasoning performance of stronger models, suggesting that correcting a few critical planning tokens can improve reasoning.

**Compliance With Llm Reviewing Policy:**

Affirmed.

**Final Justification:**

Thank authors for the detailed experiments and clarifications. My concerns have been resolved, and I would like to maintain my positive rating of this submission.

**Key Questions For Authors:**

See the weakness.

**Limitations:**

yes

**Strengths And Weaknesses:**

## Strengths

Good analysis and strategy to investigate the model reasoning process (i.e., token-level explanation of the model reasoning behavior) and identify the problem, followed by an intuitive and straightforward method.

Good writing with structured blocks of findings.


## Weaknesses

The method may increase reasoning complexity in practice, as it requires access to a stronger reasoning model to compute token-level disagreement and perform interventions. This introduces a significant dependency on the reasoning model, whose higher latency may offset the computational savings and redistribute the overall inference overhead.

Current evaluations are only conducted on math-related tasks.

The anonymous link is not accessible for reviewers to look at the detailed implementation of the method.

---

> ### Author Rebuttal · Authors · 2026-03-31
>
> We sincerely appreciate your positive feedback and valuable suggestions. We respond to your comments as follows.
>
> > **W1: The method may increase reasoning complexity in practice.**
>
> Thank you for raising this practical concern. Our main contribution is to show that the base–reasoning gap is highly sparse and concentrated in a small number of critical decision tokens. The disagreement-guided intervention is intended to validate this finding by showing that correcting only a few such tokens can recover a large fraction of reasoning performance.
>
> For practical usage, we can replace the full disagreement trigger with a cheaper uncertainty-based proxy. We therefore evaluate an entropy-triggered variant, whose results on Qwen3-0.6B are summarized below.
>
> |Method|Avg ACC|Avg Pass@8|
> |---|---:|---:|
> |Qwen3-0.6B-Base|12.9|36.0|
> |+Entropy-Guided($\bar{\rho}\approx0.08$)|65.8|87.2|
> |+Entropy-Guided($\bar{\rho}\approx0.20$)|71.3|86.4|
> |Qwen3-0.6B(Thinking)|43.4|64.1|
> |Qwen3-8B|78.1|87.3|
>
> A cheap uncertainty-based trigger already recovers most of the strong model’s Pass@8 advantage, suggesting a practical path for the same sparse-control idea.
>
> > **W2: Current evaluations are only conducted on math-related tasks**.
>
> To broaden the scope beyond math, we report performance on GPQA-Diamond [1], a general reasoning task beyond math. The same sparse intervention mechanism remains highly effective there, as summarized below.
>
> |Model|GPQA-Diamond(ACC/Pass@8)|
> |---|---:|
> |Qwen3-0.6B-Base|0.0/0.0|
> |+Guided($\bar{\rho}\approx0.01$)|18.8/82.0|
> |+Guided($\bar{\rho}\approx0.06$)|26.1/78.0|
> |+Guided($\bar{\rho}\approx0.13$)|37.1/88.0|
> |Qwen3-0.6B(Thinking)|3.5/22.0|
> |Qwen3-8B|41.5/68.0|
>
> Starting from a Qwen3-0.6B base of $0.0 / 0.0$, guided intervention substantially improves performance across all tested budgets. The best guided setting reaches $37.1 / 88.0$, exceeding the 8B model in Pass@8 ($88.0$ vs. $68.0$) while remaining close in ACC ($37.1$ vs. $41.5$). These results suggest that the sparse-intervention effect transfers beyond math.
>
> Thank you for this suggestion. We will further expand the evaluation to additional general reasoning tasks beyond math in the next version.
>
> > **W3: The anonymous code link is not accessible.**
>
> We sincerely apologize for this issue. We have re-checked and fixed the anonymous repository link, and the main implementation is now accessible.
>
> [1] Rein et al. GPQA: A Graduate-Level Google-Proof Q&A Benchmark. ICLR 2024.

---

> > ### Author Rebuttal · Reviewer_5ER5 · 2026-04-03
> >
> > Thank authors for the detailed experiments and clarifications. My concerns have been resolved, and I would like to maintain my positive rating of this submission.

---

> > > ### Author Response · Authors · 2026-04-03
> > >
> > > Thank you so much for your timely reply and positive feedback! We are glad that our response has addressed your concerns. We will incorporate all your suggestions into the revised version.

---

### Decision · Program_Chairs · 2026-04-30

**Decision:**

Accept (regular)

**Comment:**

All reviewers and myself are in consensus that this paper is deserving of acceptance. While some reviewers had concerns that the method was overly simplified and provided insufficient motivation for the design and selection of the planning tokens, in general, reviewers praised the writing of the work, straightforward methodology, and robust ablations. Furthermore, the authors presented a strong rebuttal where they addressed reviewer concerns effectively.